# Phosphorus limitation and heat stress decrease calcification in *Emiliania huxleyi*

Andrea C. Gerecht[1,a], Luka Šupraha[2,b], Gerald Langer[3], Jorijntje Henderiks[1,2]

[1]Centre for Ecological and Evolutionary Synthesis, Department of Biosciences, University of Oslo, Oslo, 0316, Norway

[2]Department of Earth Sciences, Palaeobiology, Uppsala University, Uppsala, 75236, Sweden

[3]The Marine Biological Association of the United Kingdom, The Laboratory, Citadel Hill, Plymouth, Devon, PL1 2PB, UK

[a]present address: The Faculty of Biosciences, Fisheries and Economics, UiT – The Arctic University of Norway, Tromsø, 9037, Norway

[b]present address: Section for Aquatic Biology and Toxicology, Department of Biosciences, University of Oslo, Oslo, 0316, Norway

*Correspondence to*: Andrea C. Gerecht (andrea.gerecht@uit.no)

**Abstract.** Calcifying haptophytes (coccolithophores) sequester carbon in the form of organic and inorganic cellular components (coccoliths). We examined the effect of phosphorus (P) limitation and heat stress on particulate organic and inorganic carbon (calcite) production in the coccolithophore *Emiliania huxleyi*. Both environmental stressors are related to rising $CO_2$ levels and affect carbon production in marine microalgae, which in turn impacts biogeochemical cycling. Using semi-continuous cultures, we show that P-limitation and heat stress decrease the calcification rate in *E. huxleyi*. However, using batch cultures, we show that different culturing approaches (batch versus semi-continuous) induce different physiologies. This affects the ratio of particulate inorganic (PIC) to organic carbon (POC) and complicates general predictions on the effect of P-limitation on the PIC/POC ratio. Furthermore, heat stress increases P-requirements in *E. huxleyi*, possibly leading to lower standing stocks in a warmer ocean, especially if this is linked to lower nutrient input. In summary, the predicted rise in global temperature and resulting decrease in nutrient availability may decrease $CO_2$ sequestration by *E. huxleyi* through lower overall carbon production. Additionally, the export of carbon may be diminished by a decrease in calcification and a weaker coccolith ballasting effect.

## 1 Introduction

*Emiliania huxleyi* is an abundant and ubiquitous phytoplankton species, belonging to the coccolithophores (Haptophyta), a group of calcifying microalgae. Coccolithophores fix $CO_2$ into organic matter by photosynthesis, contributing to the drawdown of atmospheric $CO_2$ (Raven and Falkowski, 1999). Calcification on the other hand, releases $CO_2$ in the short-term (Rost and Riebesell, 2004) and stores carbon in coccoliths in the long term (Sikes et al., 1980; Westbroek et al., 1993). In addition, coccolith ballast can accelerate the removal of organic carbon from upper water layers and aid long-term burial of carbon (Ziveri et al., 2007). Many studies have therefore addressed the production of organic and inorganic carbon (calcite) in *E. huxleyi*, as well as its modification by environmental factors such as carbonate chemistry (Riebesell et al., 2000; Meyer and Riebesell, 2015), nutrient availability (Paasche and Brubak, 1994; Langer and Benner, 2009), temperature (Watabe and Wilbur, 1966; Langer et al., 2010), salinity (Paasche et al., 1996; Green et al., 1998) and light (Paasche, 1968; Paasche, 1999).

This study investigates the physiological and morphological response of *E. huxleyi* to two environmental stressors, phosphorus (P) limitation and increased temperature. These are predicted to occur simultaneously as a rise in global temperature will increase the likelihood of nutrient limitation in the photic zone due to a stronger stratification of the water column (Sarmiento et al., 2004). The availability of macronutrients such as nitrogen and P have been shown to affect the production of particulate organic (POC) and inorganic carbon (PIC) in coccolithophores (reviewed by Zondervan 2007). Coccolith number per cell generally increases in P-limited cultures, often leading to an increase in the PIC/POC ratio (Paasche and Brubak, 1994; Paasche, 1998; Müller et al., 2008; Perrin et al., 2016). However, five out of six Mediterranean *E. huxleyi* strains showed a decreased PIC/POC ratio in response to P-limitation, and one strain displayed no change (Oviedo et al., 2014). While this demonstrates that there are strain-specific responses to P-limitation, some differences between studies

on PIC and POC production are due to differences in experimental methods, notably batch culture and (semi)-continuous culture approaches (Langer et al., 2013b). We used both set-ups in this study to examine the difference between strong, yet brief P-limitation (stationary phase batch culture) against weak, but continuous P-limitation (semi-continuous culture). The latter method best represents areas with permanently low nutrient availability such as the eastern Mediterranean (Krom et al., 1991; Kress et al., 2005), while stationary phase batch culture can be approximated to an end-of-bloom scenario in which the lack of nutrients limits further cell division. Both approaches are relevant in ecological terms, but for methodological reasons (i.e. non-constant growth rates), nutrient-limited production cannot be determined in the batch approach (e.g. Müller et al., 2008; Langer et al., 2012; Langer et al., 2013b, Gerecht et al., 2014; Oviedo et al., 2014; Perrin et al., 2016). In a (semi)-continuous culturing set-up, growth rate is constant over the course of the experiment and production rates can be calculated (Paasche and Brubak, 1994; Paasche, 1998; Riegman et al., 2000; Borchard et al., 2011). Ratio data such as coccolith morphology, on the other hand, should be comparable between batch and (semi)-continuous culture experiments (Langer et al., 2013b) as has been shown for *C. pelagicus* (Gerecht et al., 2014; Gerecht et al., 2015). However, the only strain of *E. huxleyi* (B92/11) that was tested in both batch and continuous culture was not analyzed for coccolith morphology and the PIC/POC ratio showed a markedly different response to P-limitation in batch and in continuous culture (Borchard et al., 2011; Langer et al., 2013b). In this study we therefore tested another strain of *E. huxleyi* in both semi-continuous and batch culture and analyzed among other things, coccolith morphology and the PIC/POC ratio.

In addition to P-limitation we studied the effect of temperature on coccolith morphology and carbon production. Only a few studies have specifically dealt with the effect of temperature on the occurrence of coccolith malformations. These studies suggest that higher than optimum temperature leads to an increase in malformations (Watabe and Wilbur, 1966; Langer et al., 2010). Although the effect of temperature on carbon production in *E. huxleyi* has been addressed in numerous studies (Sorrosa et al., 2005; Feng et al., 2008; Satoh et al., 2009; De Bodt et al., 2010; Borchard et al., 2011; Sett et al., 2014; Matson et al., 2016; Milner et al., 2016; Rosas-Navarro et al., 2016), none of these studies tested the effect of above-optimum temperature. To our knowledge, this study is the first to specifically test the impact of heat stress on carbon production in this species.

## 2 Materials & Methods

### 2.1 Cultures

We grew a strain of *E. huxleyi* isolated from the Oslo fjord (22.06.2011 by S. Ota) in triplicate semi-continuous and batch cultures in replete (control) and P-limiting medium at two temperatures (19, 24 °C). The *E. huxleyi* strain used in this study was isolated from the Oslo fjord which experiences high summer temperatures of 19-21 °C with winter lows of down to 0 °C (Aure et al., 2014). As *E. huxleyi* often has maximum growth rates at temperatures above those found at the isolation site (Sett et al., 2014), we chose 19 °C as our control temperature, which is towards the high end of the temperature range this strain is likely to encounter in nature. We used a 5 °C temperature increase to induce heat stress. This temperature (24 °C) was above the optimum for growth i.e. the cultures grew

exponentially, but at a lower rate (Eppley, 1972). Cultures were grown in modified K/2 medium (Table 1) at a salinity of 34 ppm and an initial phosphate concentration of 10 µM (control) or 0.5 µM (P-limiting). This strain belongs to morphotype A and was kept in stock culture at 12 °C under low light in K/2 medium prior to the start of the experiment. The strain has been deposited at the NIVA Culture Collection of Algae (niva-cca.no) as strain UIO 265. The cultures were grown in an environmental test chamber (MLR-350, Panasonic, Japan), on a 12:12 h light:dark cycle at an irradiance of ~100 µmol photons $m^{-2}$ $s^{-1}$. Cultures were acclimated for ca. 10 generations to the two initial P-concentrations and temperatures before starting the experiment.

Semi-continuous cultures were inoculated at an initial cell concentration of ~10.000 cells $mL^{-1}$ and diluted back to this cell concentration with fresh medium every second day. Cell concentrations were determined daily using an electronic particle counter (CASY, Roche Diagnostics, Switzerland). Maximum cell concentrations (<170.000 cells $mL^{-1}$) were well below stationary phase so that all cultures were kept continuously in the exponential growth phase (supplementary Fig. 1). Semi-continuous cultures were harvested after 10 dilution cycles. For batch cultures, the initial inoculum was also ~10.000 cells $mL^{-1}$. P-limited cultures were harvested in stationary phase, whereas control cultures were harvested in exponential phase at similar cell concentrations (see Gerecht et al., 2014; Fig. 1).

Exponential growth rates ($\mu_{exp}$) were calculated by linear regression of log-transformed cell concentrations over time. For batch cultures, only the exponential part of the growth curve was considered. For semi-continuous cultures, $\mu_{exp}$ was calculated as an average of $\mu_{exp}$ of all dilution cycles.

## 2.2 Medium chemistry

### 2.2.1 Phosphate concentrations

Residual phosphate concentrations were determined in the culture media upon harvest of the cultures. The medium was sterile filtered (0.2 µm) into plastic scintillation vials (Kartell, Germany) and stored at -20 °C until analysis. Phosphate concentrations were determined colorimetrically on a spectrophotometer (UV 2550, Shimadzu, Japan) as molybdate reactive phosphate following Murphy and Riley (1962) with a precision of ± 4 %.

### 2.2.2 Carbonate chemistry

Total alkalinity ($A_T$) and pH of the medium were determined upon harvest of the cultures. The initial carbonate chemistry of the culture media is presented in Table 1. Samples for $A_T$ were filtered through GF/F-filters (Whatman, GE Healthcare, UK), stored airtight at 4 °C and analyzed within 24 h. $A_T$ was calculated from Gran plots (Gran, 1952) after duplicate manual titration with a precision of ± 50 µmol $kg^{-1}$. The pH was measured with a combined electrode (Red Rod, Radiometer, Denmark) that was two-point calibrated to NBS scale (precision ± 0.03). Dissolved inorganic carbon (DIC) concentrations and saturation state of calcite ($\Omega_{Ca}$) were calculated using CO2sys (version 2.1 developed for MS Excel by D. Pierrot from E. Lewis and D. W. R. Wallace) using $A_T$ and pH as input parameters and the dissociation constants for carbonic acid of Roy et al. (1993).

### 2.3 Elemental composition

#### 2.3.1 Particulate organic phosphorus

Samples for particulate organic phosphorus (POP) were filtered onto precombusted (500 °C, 2 h) GF/C-filters (Whatman) and stored at -20 °C. POP was converted to orthophosphate by oxidative hydrolysis with potassium persulfate under high pressure and temperature in an autoclave (3150EL, Tuttnauer, Netherlands) according to Menzel and Corwin (1965). Converted orthophosphate was then quantified as molybdate reactive phosphate as described in Sect. 2.2.1.

#### 2.3.2 Particulate organic and inorganic carbon

Samples for total particulate carbon (TPC) and particulate organic carbon (POC) were filtered onto precombusted GF/C-filters, dried at 60 °C overnight in a drying oven, and stored in a desiccator until analysis on an elemental analyzer (Flash 1112, Thermo Finnegan, USA; detection limit 2 µg; precision ± 8 %). Particulate inorganic carbon (PIC) was removed from POC filters by pipetting 230 µL of 2 M HCl onto the filters before analysis (Langer et al., 2009) and calculated as the difference between TPC and POC.

### 2.4 Cell geometry

Cell volume was calculated from cell diameters measured both visually from light microscopy images (LM) and automatically with an electronic particle counter (CASY). With LM, cell diameters of live cells were measured at 200 times magnification after dissolving the coccoliths with 0.1 M HCl (19 µL to 1 mL sample; Gerecht et al., 2014) after harvesting the cultures. CASY cell diameters were recorded during daily measurements of cell concentrations (see 2.1) without removing coccoliths. Cell volume derived from CASY data therefore overestimates actual cell volume, because part of the coccosphere is included. However, volume estimates from CASY data are based on the measurement of many cells, leading to robust data i.e. a low standard deviation compared to LM measurements (Table 2). They are therefore useful for comparative purposes and for following the development of cell size during culture growth (Fig. 1; see also Gerecht et al., 2015).

A Zeiss Supra35-VP field emission scanning electron microscope (Zeiss, Germany) was used to capture images for morphological analyses. The number of coccoliths per coccosphere was estimated from these images by counting visible, forwards facing coccoliths, multiplying this number by two to account for the coccoliths on the back side of the coccosphere, and adding the number of partially visible coccoliths along its edge (Gerecht et al., 2015). Coccolith morphology was classified into three categories: normal, incomplete, and malformed (Table 3; Fig. 2). Due to the low calcite saturation state reached in stationary phase batch cultures, we observed a high number of partially dissolved coccoliths in these cultures (the features of this secondary dissolution are described in Table 3 and Fig. 2). As it was not possible to unambiguously distinguish incomplete morphology due to secondary dissolution from incompletely produced coccoliths, only one class of incomplete coccoliths is presented in Fig. 3.

### 2.5 Statistical treatment of the data

The average value of parameters from triplicate cultures is given as the statistical mean together with standard deviation. The influence of P-availability and temperature on variables was determined by means of a two-way analysis of variance (ANOVA). For discrete data (DIC, coccolith morphology), a non-parametric test (Mann-Whitney U test) was used. All statistical treatment of the data was preformed using Statistica (release 7) software (StatSoft, USA).

## 3 Results

### 3.1 Semi-continuous cultures

Particulate organic phosphorus (POP) cellular content (F-value=24.46, $p<0.001$) and production (F-value=20.92, $p<0.001$) were significantly lower in P-limited than in control cultures (Table 4; supplementary Fig. 2). P-limitation, however, had no effect on exponential growth rate ($\mu_{exp}$) (F-value=0.54, $p=0.47$), POC content (F-value=4.16, $p=0.055$), POC production (F-value=3.71, $p=0.09$) or cell size (Table 2; F-value=0.21, $p=0.65$). PIC production, on the other hand, was significantly decreased in P-limited cultures (Table 4; F-value=13.25, $p=0.0066$) and P-limited cells were covered by one to two fewer coccoliths (Table 2; Fig. 4a,b), which led to a decrease in the PIC/POC ratio (Table 4; F-value=19.01, $p=0.0024$). Coccolith morphology was unaffected by P-limitation (Table 2, Fig. 3; Z-value=-0.40, $p=0.69$).

The 5 °C temperature increase from 19 to 24 °C decreased $\mu_{exp}$ by 10 % in control cultures and by 7 % in P-limited cultures (Table 4; F-value=20.74, $p<0.001$). POC production, however, was unaffected by temperature (F-value=0.38, $p=0.55$) as there was a significant increase in POC content (supplementary Fig. 2; F-value=8.52, $p=0.0085$) and cell size (Table 2; F-value=10.36, $p=0.0029$) at 24 °C. PIC production was significantly lower at 24 °C (Table 4; F-value=19.73, $p=0.0022$) and the cells were covered by three to four fewer coccoliths (Table 2; Fig. 4b). The lowest PIC/POC ratio (0.81 ± 0.06) and coccolith numbers per cell (15 ± 5) were therefore observed in P-limited cultures at 24 °C. There was a strong increase in the occurrence of malformed coccoliths at 24 compared to 19 °C (Table 2; Z-value=-2.88, $p=0.0039$).

There was no direct effect of temperature on POP content (Table 4; supplementary Fig. 2; F-value=2.66, $p=0.12$). There was, however, a combined effect of temperature and P-limitation (F-value=4.49, $p=0.047$) so that the lowest POP content was measured in P-limited cultures at 24 °C. These cultures had taken up most of the phosphate from the medium by the time of harvest (Table 5).

### 3.2 Batch cultures

Cells from control batch cultures were overall smaller than those from semi-continuous cultures (Table 2) and consequently contained less POP and POC (Table 4; supplementary Fig. 2). POC/POP-values of control batch and control semi-continuous cultures, however, were similar.

Initial phosphate availability did not affect $\mu_{exp}$ (F-value=3.19, $p=0.11$). At 19 °C, cultures growing in P-limiting medium stopped dividing at a cell concentration of ~740,000 cells mL$^{-1}$. At 24 °C, final cell concentrations in stationary phase were significantly lower at ~620,000 cells mL$^{-1}$ (t-value=13.77, $df=16$, $p<0.001$). Final DIC concentrations were significantly lower at 19 °C (400 ± 50 µmol kg$^{-1}$) than at 24 °C (550 ± 50 µmol kg$^{-1}$; Table 5; Z-value=-2.62, $p<0.01$), whereas DIC concentrations remained

at ~1000 µmol kg$^{-1}$ in control cultures. The pH of the culture medium in P-limited batch cultures was also significantly different between the two temperatures. At 19 °C, the final pH-value was 7.70 ± 0.02 compared to 7.85 ± 0.01 at 24 °C. In control cultures, the pH stayed close to normal seawater values (~8.2) at both temperatures. P-limited cultures were undersaturated in calcite ($\Omega_{Ca}$<1) at the time of harvest with a significantly stronger undersaturation at 19 °C ($\Omega_{Ca}$ = 0.40 ± 0.03) than at 24 °C ($\Omega_{Ca}$ = 0.77 ± 0.05; Z-value=-2.61, $p$<0.01).

POP content was ~3-4 times lower at both temperatures in P-limited than in control cultures (Table 4; supplementary Fig. 2). However, POP content was significantly higher in cultures grown at 24 °C (83 ± 3 ng cell$^{-1}$) than at 19 °C (71 ± 9 ng cell$^{-1}$; t-value=-3.24, $df$=10, $p$<0.01). Cells from P-limited cultures increased in size as cell division rates slowed down (Fig. 1) and cell volume was twice as large in stationary phase than in control cultures in exponential phase (Table 2, Fig. 4c,d). This coincided with a 2.7- and 2.1-fold increase in POC content in P-limited cultures at 19 and 24 °C, respectively (Table 4; supplementary Fig. 2).

In P-limited cultures, the average number of coccoliths per cell tripled at 19 °C (from ~15 to ~45 coccoliths cell$^{-1}$) and doubled (from ~16 to ~34 coccoliths cell$^{-1}$) at 24 °C (Table 2). The PIC content, on the other hand, increased by ~150 % at both temperatures (Table 4; Fig. 4c,d). Coccolith morphology was obscured in P-limited cultures by secondary dissolution with 77 % of all coccoliths showing incomplete morphology at 19 °C and 52 % of coccoliths at 24 °C (Table 2; Fig. 3). The percentage of incomplete coccoliths was negligible in control cultures. Coccolith malformations were twice as common in control cultures at 24 °C than at 19 °C (Table 2; Fig. 3; Z-value=-1.96, $p$=0.049). Temperature had no effect on $\mu_{exp}$ (F-value=3.19, $p$=0.11) or on production rates in control cultures (Table 4).

**4 Discussion**

**4.1 The effect of P-limitation on PIC and POC production**

When testing nutrient limitation in a laboratory setting, it is important to consider the putative physiological difference between cells growing exponentially at lower nutrient availability (continuous or semi-continuous culture) and cells entering stationary phase once the limiting nutrient has been consumed (batch culture) (Langer et al., 2013b; Gerecht et al., 2015). While the former allows for acclimation to lower nutrient availability, the latter creates a strong limitation of short duration that leads to a cessation of cell division. A good parameter to assess this potential physiological difference is the PIC/POC ratio, because, in contrast to PIC and POC production, it can be determined in both batch and continuous culture (Langer et al., 2013b). Despite the considerable body of literature on carbon production under P-limitation in *E. huxleyi* (see Introduction), only one strain (B92/11) has been examined in a comparative study showing that the PIC/POC response to P-limitation varies with the approach chosen (Borchard et al., 2011; Langer et al., 2013b). The case of *E. huxleyi* B92/11 suggests that the physiological state induced by P-limitation in batch culture indeed differs from the one induced by P-limitation in continuous culture. In this strain P-limitation decreased the PIC/POC ratio in batch culture (Langer et al., 2013b), while no change occurred in continuous culture (Borchard et al., 2011). In the strain used in this study the opposite is true, i.e. the PIC/POC ratio decreased in

semi-continuous culture and remained constant in batch culture at normal temperature. The highly variable PIC/POC response to P-limitation observed here and in B92/11 (Borchard et al., 2011; Langer et al., 2013b) shows that the physiological state under P-limitation depends on the experimental approach, and that there is no clear trend in the response pattern among different strains. Consequently it is difficult to formulate a common scenario with respect to carbon allocation under P-limitation. However, our semi-continuous culture experiment shows that in this strain under P-limitation POC production remains unchanged and PIC production decreases. The 14 % decrease in PIC production observed here is quite remarkable, because the limitation imposed by our semi-continuous setup was weak as can be inferred from the maintained growth rate and the weak (11 %) decrease in POP production. Hence in this strain of *E. huxleyi* the calcification rate is particularly sensitive to P-limitation. As this is the first report of P-limitation decreasing coccolith production in *E. huxleyi*, it would be beneficial to test further strains in a similar set-up to observe how common this physiological response is in *E. huxleyi*. Ecological benefits of coccoliths are likely to be various (Monteiro et al., 2016). Protection from UV-radiation (Xu et al., 2011), for example, may be relevant as this species grows at high light intensities. Furthermore, the consumption of coccoliths by grazers in addition to organic cell material may decrease overall grazing rates (Monteiro et al., 2016). A decrease in coccolith coverage may therefore constitute a loss in overall fitness of an *E. huxleyi* population. Coccolith morphogenesis, on the other hand, was unaffected by P-limitation. This reflects the potentially wide spread insensitivity of coccolith morphogenesis to P-limitation (Langer et al., 2012; Oviedo et al., 2014) with the exception of *C. pelagicus* (Gerecht et al. 2015).

In a recent study, Bach et al. (2013) determined that POC production in *E. huxleyi* is DIC-limited at concentrations <1000 µmol $kg^{-1}$. Final DIC concentrations in our stationary phase cultures were well below that value and these cultures were possibly limited in both P and DIC at the time of harvest. DIC-limitation, however, was not the trigger for entering stationary phase as POC production continued for several days after cessation of cell division. Wördenweber et al. (2017) have recently shown that although the cell cycle is arrested by P-starvation, enzymatic functionality is widely preserved. P-starvation blocks the synthesis of DNA and membrane phospholipids, necessary for cell replication, arresting the cells in the G1 (assimilation) phase of the cell cycle (Müller et al., 2008). The assimilation phase is thus prolonged and the cell continues assimilating POC, presumably in the form of non-essential lipids and carbohydrates (Sheward et al., 2017), leading to an increase in cell size (Aloisi, 2015). A similar increase in cell size as observed in this study has been previously described by others for *E. huxleyi* (Paasche and Brubak, 1994; Riegman et al., 2000; Müller et al., 2008; Gibbs et al., 2013; Oviedo et al., 2014) and recently also for other species, such as *C. pelagicus*, *Helicosphaera carteri* and two *Calcidiscus* species (Gerecht et al., 2015; Sheward et al., 2017) and may thus be a common feature of coccolithophores.

Cells that are arrested in the G1 (assimilation) phase of the cell cycle (Gibbs et al., 2013), not only accumulate POC, but also accumulate PIC, leading to the 2-3-fold increase in coccolith number per cell observed in stationary phase cultures (Fig. 4c,d). Stationary phase can be likened to an end-of-bloom scenario in nature, during which *E. huxleyi* sheds numerous coccoliths, leading to the characteristic milky color of coccolithophore blooms (Balch et al., 1991; Holligan et al., 1993). Though these blooms

are important contributors to the sequestration of atmospheric $CO_2$ and carbon export, they are short-lived phenomena. The present data set is unique in providing information on PIC production under P-limitation without the confounding factor of changes in growth rate. By using semi-continuous cultures in which cell division rates remained constant between control and P-limited cultures, we could show that the likely outcome of diminished P-availability will be a long-term decrease in PIC production in *E. huxleyi*, which may weaken carbon export from surface waters (Ziveri et al., 2007).

### 4.2 The effect of heat stress on calcification

The decrease in growth rate at 24 °C, observed in semi-continuous cultures, confirmed that this temperature was indeed above the optimum for growth for this particular strain (Eppley, 1972). Although a similar decrease in growth rate was not observed in batch culture, measurements of growth rate in semi-continuous cultures are more robust because growth rate is measured as an average of numerous dilution cycles. The doubling in coccolith malformations provides further evidence that 24 °C cultures were indeed heat-stressed (Watabe and Wilbur, 1966; Langer et al., 2010; Milner et al., 2016).

The POP content of (P-limited) stationary phase cultures can be used as an indicator for minimum P-requirements (Šupraha et al., 2015). These increased by ~17 % under heat stress. Increased P-requirements led to lower final biomass, both in terms of final cell numbers and lower cellular POC content in heat-stressed cultures. An increase in P-requirements at higher temperature has previously been described for the coccolithophore *C. pelagicus*, which also led to lower final cell numbers in stationary phase P-limited cultures (Gerecht et al., 2014). Higher P-requirements at higher temperature can be furthermore inferred for two additional strains of *E. huxleyi* from the studies carried out by Feng et al. (2008) and Satoh et al. (2009). Increased P-requirements at higher temperature may therefore be a general feature of coccolithophores with the potential to decrease coccolithophore carbon production in a future warmer ocean. A similar increase was not observed in heat-stressed, exponentially growing cultures i.e. control batch and semi-continuous cultures because P-uptake was 3-4 times higher than the minimum requirement. The low residual phosphate concentrations of P-limited semi-continuous cultures are also indicative of increased P-uptake under heat stress. This was not reflected in the POP content, which was actually lower under heat stress. A possible explanation for these conflicting results may be an increased production of exudates due to heat stress with a concomitant loss of organic P from the cell (Borchard and Engel, 2012). Higher P-requirements may be due either to increased energy demands under heat stress or to an upregulation of heat stress related genes as much of cellular P can be found in RNA (Geider and LaRoche, 2002).

Heat stress had a stronger effect than P-limitation on coccolith number in semi-continuous cultures. Whereas P-limited cells were covered by one to two fewer coccoliths, heat stress decreased the number of coccoliths per cell by three to four coccoliths (Fig. 4a,b). In *C. pelagicus*, heat stress has also been described to decrease the coccolith coverage of the cell (Gerecht et al., 2014). Also in P-limited batch cultures, fewer coccoliths accumulated around the cells under heat stress (Fig. 4d). This was not, however, reflected by a lower PIC content of these cells. There are two possible mechanisms to explain this incongruence. One reason may be the partial dissolution of coccoliths in P-limited stationary phase

cultures. High numbers of partially dissolved coccoliths were observed in P-limited batch cultures at both temperatures due to the low calcite saturation state reached in stationary phase cultures. However, the occurrence of secondary dissolution was higher at normal temperature than under heat stress as these cultures reached higher final biomass and consequently were less saturated in calcite. These partially dissolved coccoliths likely contained less calcite, which may explain why the cellular PIC content was similar at both temperatures even if the coccolith number per cell differed. Due to this secondary dissolution, the PIC quota and PIC/POC ratio measured in P-limited batch cultures are most likely underestimated, especially at normal temperature, and need to be interpreted with caution.

Another possible reason for the discrepancy between PIC and coccolith quota between the two temperatures is a difference in the ratio of attached to loose coccoliths. Possibly, more coccoliths were shed under heat stress, underestimating the coccolith number of these cells. As *E. huxleyi* in general sheds many coccoliths, this effect can be considerable (Milner et al., 2016). We therefore cannot conclusively determine whether the effect of P-limitation on the PIC/POC ratio was modified by heat stress in batch culture. Despite the high percentage of partially dissolved coccoliths in P-limited batch culture, the detrimental effect of heat stress on morphogenesis is evident. As all *E. huxleyi* strains tested so far show this response, it could be widespread if not ubiquitous (Watabe and Wilbur, 1966; Langer et al., 2010, this study). Interestingly, we observed similar malformations e.g. merged distal shield elements in field samples collected from the Oslo fjord (Fig. 2d) at a time when *E. huxylei* was abundant in the water column (Gran-Stadniczeňko et al., 2017). The percentage of malformed coccoliths in field samples was lower (ca. 6%) than in our control cultures (ca. 20%), lending support to the hypothesis that coccolith malformations occur more frequently in culture (Langer et al., 2013a). The types of malformations, however, appear to be similar, indicating that the affected physiological mechanisms are the same.

De Bodt et al. (2010) described a decrease in the PIC/POC ratio at higher temperature in *E. huxleyi*. Several studies have contrastingly reported the PIC/POC ratio to be insensitive to temperature (Feng et al., 2008; Matson et al., 2016; Milner et al., 2016) or to increase with rising temperatures (Sett et al., 2014). In all of the above studies, however, growth rate increased from low to high temperature and none of the tested temperatures were therefore above the optimum for growth (Eppley, 1972). To our knowledge, this study is the first to show that heat stress is not only detrimental for coccolith morphology (Watabe and Wilbur, 1966; Langer et al., 2010; Milner et al., 2016), but also for coccolith production in *E. huxleyi*. Certainly, the potential for long-term adaptation needs to be considered, as temperature increases are unlikely to occur on time scales short enough to preclude adaptation in a rapidly growing species. The species *E. huxleyi* is present also at higher temperatures in nature (Feng et al., 2008) so a physiological constraint to adaptation to higher temperatures is not probable. Similarly, considering the metabolic diversity among different *E. huxleyi* strains (Langer et al., 2009; Read et al., 2013), this strain could be replaced by a more heat-tolerant strain.

5. **Conclusions**

By employing semi-continuous cultures, we show that both P-limitation and heat stress decrease calcification rate in a temperate strain of *E. huxleyi*. Considering that these stressors are likely to co-

occur in a future ocean (Sarmiento et al., 2004), it is important to consider this additive effect. The increase in cellular P-requirements under heat stress may intensify nutrient limitation, decreasing the standing stock of *E. huxleyi* in a warmer ocean, which would have a negative feedback on carbon sequestration. An increase in P-requirements and decrease in coccolith production under heat stress have also been described for *C. pelagicus* and may be a general feature of coccolithophores. To what extent a decrease in calcification under weak P-limitation is a general feature of *E. huxleyi* needs to be verified by additional studies, considering that the response of the PIC/POC ratio to P-limitation is both strain and method dependent. The method dependency is due to the determining effect of cell size and cell division rate i.e. growth phase on the PIC/POC ratio. This high variability of the PIC/POC ratio, one of the most important parameters in biogeochemical terms, makes it difficult to predict the impact of P-limitation in *E. huxleyi* on the carbon cycle. However, we have shown that lower phosphorus input and higher global temperature can have an additive negative effect on calcification. Decreased calcification rates weaken carbon export due to less coccolith ballasting (Ziveri et al., 2007). It is therefore fundamental to understand how environmental factors interact in their effect on calcification in coccolithophores – from the cellular to the ecological level.

## Author contributions

AG, JH, and GL designed the experiments. AG and LS carried out the experiment. All authors interpreted the findings. AG prepared the manuscript with contribution from all co-authors.

The authors declare that they have no conflict of interest.

## Acknowledgments

This research was funded by the Research Council of Norway (FRIMEDBIO project 197823 to JH) and the Royal Swedish Academy of Sciences through a grant from the Knut and Alice Wallenberg Foundation (KAW 2009.0287 to JH). The authors would like to acknowledge T. Claybourn (Department of Earth Sciences, Palaeobiology, Uppsala University) who collected the morphology images using scanning electron microscopy. The authors would furthermore like to thank Prof. B. Edvardsen (Section for Aquatic Biology and Toxicology, Department of Biosciences, University of Oslo) for fruitful discussions of the presented data.

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

Table 1: Basal composition of the culture media (modified K/2), including salinity and carbonate chemistry (total alkalinity, pH, dissolved inorganic carbon, calcite saturation state). * For trace metal composition please refer to the recipe available for K/2 Ian at roscoff-culture-collection.org/basic-page/culture media.

| Final conc. [μM] | "Control, replete medium" | "P-limited medium" |
| --- | --- | --- |
| NaNO$_3$ | 288 | 288 |
| KH$_2$PO$_4$ | 10 | 0.5 |
| (Na)FeEDTA | 5.85 | 5.85 |
| Trace metals | * | * |
| Vitamins | "f/2" | "f/2" |
| Salinity [ppm] | 34 | 34 |
| $A_T$ [μmol kg$^{-1}$] | 2250 | 2100 |
| pH (NBS) | 8.14 | 7.89 |
| DIC [μmol kg$^{-1}$] | 1800 | 1800 |
| $\Omega_{Ca}$ | 4.7 | 2.7 |

Table 2: Cell volume calculated from light microscopy (LM) and electronic particle counter (CASY) measurements at the time of harvest, number of coccoliths cell$^{-1}$ and number of coccoliths analyzed by scanning electron microscopy (SEM) and classified into normal, incomplete and malformed coccoliths in semi-continuous and batch control and P-limited cultures of *Emiliania huxleyi* grown at 19 and 24

5 °C. The number of cells (n) analysed for each measurement is presented as the sum of three replicates, except for CASY cell volume measurements for which n=3; ± standard deviation.

| | Semi-continuous cultures | | Batch cultures | |
|---|---|---|---|---|
| | Control | P-limited | Control | P-limited |
| Cell volume [μm$^3$] (LM) | | | | 10 |
| 19 °C | 34.3 ± 17.7 (n=111) | 19.9 ± 9.5 (n=116) | 29.7 ± 12.1 (n=346) | 57.6 ± 22.7 (n=205) |
| 24 °C | 24.6 ± 12.8 (n=117) | 34.3 ± 17.7 (n=194) | 24.7 ± 14.1 (n=352) | 64.3 ± 31.7 (n=217) 15 |
| (CASY) | | | | |
| 19 °C | 74.4 ± 8.9 | 75.5 ± 7.4 | 62.7 ± 7.3 | 106.9 ± 9.8 |
| 24 °C | 94.0 ± 4.5 | 92.1 ± 6.8 | 67.1 ± 5.8 | 115.1 ± 3.0 20 |
| Coccoliths cell$^{-1}$ | | | | |
| 19 °C | 20 ± 9 (n=148) | 18 ± 6 (n=149) | 15 ± 5 (n=151) | 45 ± 20 (n=149) |
| 24 °C | 16 ± 7 (n=145) | 15 ± 5 (n=146) | 16 ± 6 (n=149) | 34 ± 15 (n=145) 25 |
| Number of coccoliths analysed for morphology | | | | |
| 19 °C | 821 | 824 | 693 | 3496 |
| 24 °C | 731 | 721 | 691 | 2010 30 |
| Normal [%] | | | | |
| 19 °C | 81.5 | 79.5 | 77.6 | 21.3 |
| 24 °C | 51.0 | 54.7 | 57.1 | 33.8 |
| Incomplete [%] | | | | |
| 19 °C | 1.3 | 0.8 | 1.8 | 76.7 |
| 24 °C | 2.0 | 0.7 | 4.4 | 52.4 |
| Malformed [%] | | | | |
| 19 °C | 17.2 | 19.7 | 20.6 | 2.0 |
| 24 °C | 46.9 | 44.7 | 38.5 | 13.7 |

Table 3: Characteristics of the three classes (normal, incomplete, malformed) used to describe coccolith morphology, including a description of "dissolution features"

| Coccolith type | Description |
| --- | --- |
| Normal | Central area, proximal and distal shield fully developed; distal shield elements clearly separated by slits with complete outer rim of the distal shield (Fig. 2a). |
| Incomplete | Central area, proximal and/or distal shield not fully developed; incomplete or absent outer rim of the distal shield (Fig. 2b), but without visible malformations of distal shield elements (as defined below). |
| Malformed | Several types of malformations were observed (Fig. 2c-g): 1) more than two merged distal shield elements (Fig. 2c) 2) tips of distal shield elements forming triangular thickening with outer rim (Fig. 2e) 3) increased gaps between distal shield elements (Fig. 2c) 4) missing central area (Fig. 2d) 5) irregular outgrowth of calcite (Fig. 2e) 6) strongly malformed coccoliths of irregular shape (Fig. 2e) |
| Signs of secondary dissolution | Distal shield elements thinning or detaching (Fig. 2f, g); incomplete outer rim with "hammer-like" distal shield elements (Fig. 2f); thinning central area (Fig. 2g); thinning of the proximal shield with exposed shield elements separated by slits (Fig. 2g); coccoliths lose their structural integrity and coccospheres are mostly collapsed (Fig. 2g). |

Table 4: Exponential growth rate ($\mu_{exp}$), particulate organic phosphorus (POP), carbon (POC) and inorganic carbon (PIC) cellular content, production and ratios in semi-continuous and batch control and P-limited cultures of *Emiliania huxleyi* grown at 19 and 24 °C; n=3 ± standard deviation.

| | Semi-continuous cultures | | Batch cultures | |
|---|---|---|---|---|
| | Control | P-limited | Control | P-limited |
| $\mu_{exp}$ | | | | |
| 19 °C | 1.32 ± 0.05 | 1.31 ± 0.02 | 1.08 ± 0.07 | 1.15 ± 0.03 |
| 24 °C | 1.20 ± 0.07 | 1.23 ± 0.07 | 1.15 ± 0.02 | 1.18 ± 0.04 |
| POP [pg cell$^{-1}$] | | | | |
| 19 °C | 0.42 ± 0.03 | 0.38 ± 0.03 | 0.26 ± 0.03 | 0.071 ± 0.009 |
| 24 °C | 0.43 ± 0.03 | 0.33 ± 0.05 | 0.27 ± 0.02 | 0.083 ± 0.003 |
| POP [pg cell$^{-1}$ d$^{-1}$] | | | | |
| 19 °C | 0.56 ± 0.04 | 0.50 ± 0.04 | 0.28 ± 0.02 | n/a |
| 24 °C | 0.51 ± 0.03 | 0.40 ± 0.06 | 0.32 ± 0.02 | n/a |
| POC [pg cell$^{-1}$] | | | | |
| 19 °C | 13.5 ± 0.9 | 14.8 ± 0.7 | 8.1 ± 0.7 | 21.5 ± 0.8 |
| 24 °C | 15.1 ± 1.2 | 15.3 ± 0.5 | 8.9 ± 0.3 | 18.3 ± 0.4 |
| POC [pg cell$^{-1}$ d$^{-1}$] | | | | |
| 19 °C | 17.8 ± 1.2 | 19.3 ± 1.0 | 8.8 ± 0.4 | n/a |
| 24 °C | 18.1 ± 1.4 | 18.8 ± 0.6 | 10.5 ± 0.1 | n/a |
| POC/POP [mol mol$^{-1}$] | | | | |
| 19 °C | 82.8 ± 5.2 | 101 ± 8 | 79.9 ± 1.8 | 792 ± 93 |
| 24 °C | 91.1 ± 7.0 | 123 ± 16 | 85.3 ± 6.9 | 572 ± 17 |
| PIC [pg cell$^{-1}$] | | | | |
| 19 °C | 14.7 ± 0.9 | 12.8 ± 0.6 | 6.6 ± 0.6 | 16.5 ± 0.4[a] |
| 24 °C | 13.6 ± 1.3 | 12.4 ± 0.7 | 7.3 ± 0.3 | 18.7 ± 0.9[a] |
| PIC [pg cell$^{-1}$ d$^{-1}$] | | | | |
| 19 °C | 19.4 ± 1.2 | 16.7 ± 0.8 | 7.1 ± 0.3 | n/a |
| 24 °C | 16.3 ± 1.5 | 15.3 ± 0.9 | 8.6 ± 0.3 | n/a |
| PIC/POC | | | | |
| 19 °C | 1.09 ± 0.07 | 0.87 ± 0.07 | 0.81 ± 0.03 | 0.77 ± 0.02[a] |
| 24 °C | 0.90 ± 0.08 | 0.81 ± 0.06 | 0.82 ± 0.03 | 1.02 ± 0.04[a] |

[a]presumably underestimated because of calcite undersaturation (see Table 5)

Table 5: Cell concentrations, residual phosphate, total alkalinity, pH, dissolved inorganic carbon, and saturation state of calcite in the culture medium at the time of harvest of semi-continuous and batch control and P-limited cultures of *Emiliania huxleyi* grown at 19 and 24 °C; n=3 ± standard deviation.

| | Semi-continuous cultures | | Batch cultures | |
|---|---|---|---|---|
| | Control | P-limited | Control | P-limited |
| $\times 10^4$ cells mL$^{-1}$ | | | | |
| 19 °C | 8.29 ± 0.54 | 7.87 ± 0.46 | 78.32 ± 16.38 | 73.78 ± 2.26 |
| 24 °C | 14.26 ± 1.50 | 14.98 ± 0.66 | 79.99 ± 1.16 | 61.63 ± 1.37 |
| $PO_4^{3-}$ [µM] | | | | |
| 19 °C | 6.41 ± 0.37 | 0.50 ± 0.05 | 3.58 ± 1.03 | 0.18 ± 0.09 |
| 24 °C | 6.65 ± 0.95 | 0.06 ± 0.03 | 2.93 ± 0.39 | 0.06 ± 0.04 |
| $A_T$ [µmol kg$^{-1}$] | | | | |
| 19 °C | 2000 ± 50 | 2100 ± 50 | 1450 ± 100 | 500 ± 50 |
| 24 °C | 1950 ± 50 | 2000 ± 50 | 1250 ± 50 | 700 ± 50 |
| pH (NBS) | | | | |
| 19 °C | 8.01 ± 0.01 | 8.05 ± 0.04 | 8.21 ± 0.06 | 7.70 ± 0.02 |
| 24 °C | 8.13 ± 0.06 | 8.16 ± 0.11 | 8.22 ± 0.02 | 7.85 ± 0.01 |
| DIC [µmol kg$^{-1}$] | | | | |
| 19 °C | 1650 ± 50 | 1700 ± 50 | 1050 ± 100 | 400 ± 50 |
| 24 °C | 1550 ± 100 | 1550 ± 100 | 950 ± 50 | 550 ± 50 |
| $\Omega_{Ca}$ | | | | |
| 19 °C | 3.14 ± 0.08 | 3.66 ± 0.29 | 3.20 ± 0.16 | 0.40 ± 0.03 |
| 24 °C | 3.91 ± 0.35 | 4.38 ± 0.69 | 2.93 ± 0.10 | 0.77 ± 0.05 |

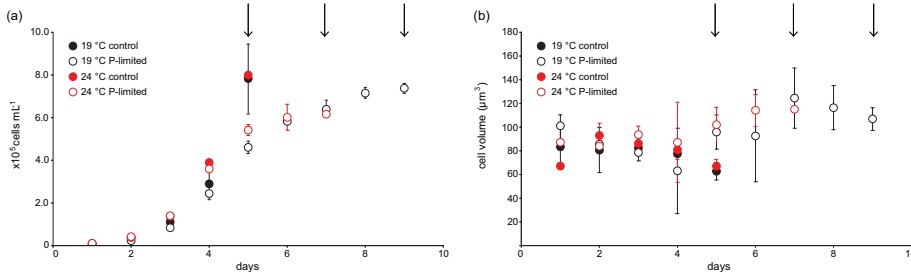

**Figure 1: (a) Cell concentrations and (b) cell volume over time in batch cultures of *Emiliania huxleyi* grown at 19 and 24 °C in control and P-limited medium. Error bars denote the standard deviation of mean triplicate measurements of triplicate cultures. Arrows indicate when cultures were harvested.**

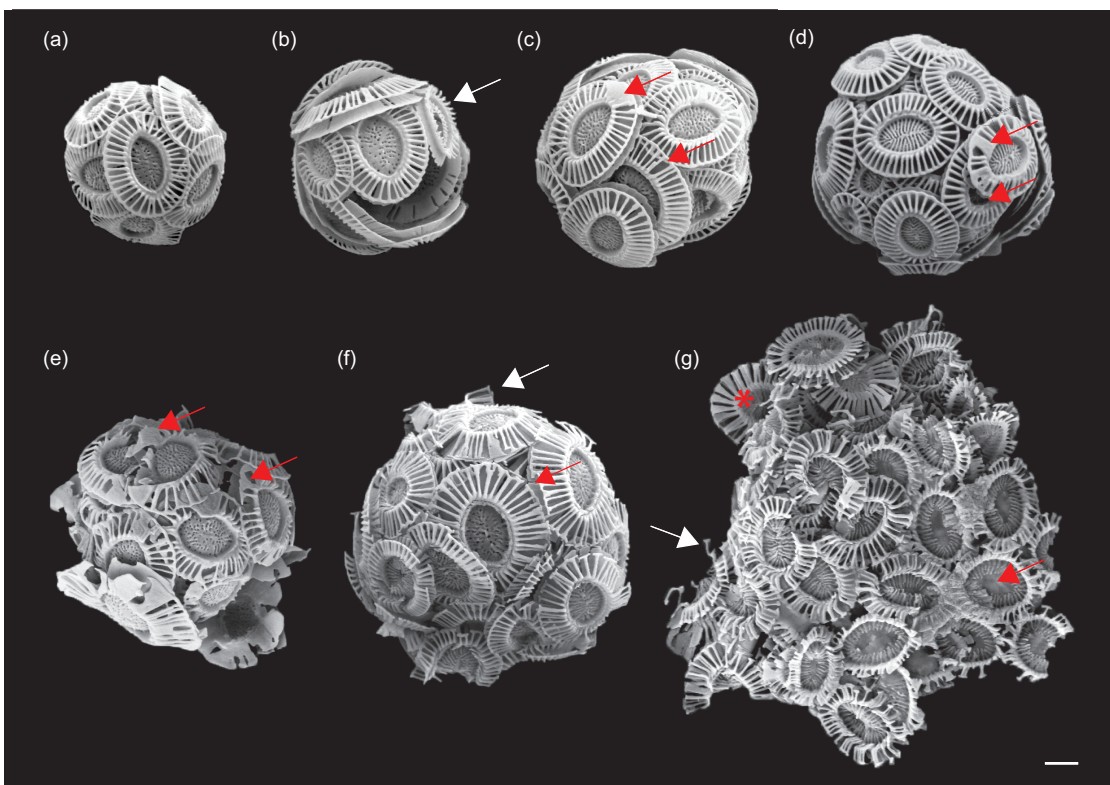

**Figure 2: Representative scanning electron micrographs of normal, incomplete and malformed coccoliths, including dissolution features: (a) coccosphere bearing normal coccoliths; (b) arrow: an incomplete coccolith in control batch culture; (c) arrows: malformed coccoliths with merged distal shield elements/increased gaps (d) a coccosphere from an Oslo fjord field sample; arrows highlight the same type of malformations (merged distal shield elements, missing central area) as observed in culture; (e) coccosphere with many malformed coccoliths showing merged distal shield elements, triangular thickening of the elements and irregular calcite growth; (f) partially dissolved coccosphere; white arrow: detached distal shield elements; red arrow: "hammer-like" distal shield elements; (g) strongly dissolved coccosphere; white arrow: detached distal shield elements; red arrow: dissolved central area; asterisk: exposed proximal shield elements. Scale bar = 1µm.**

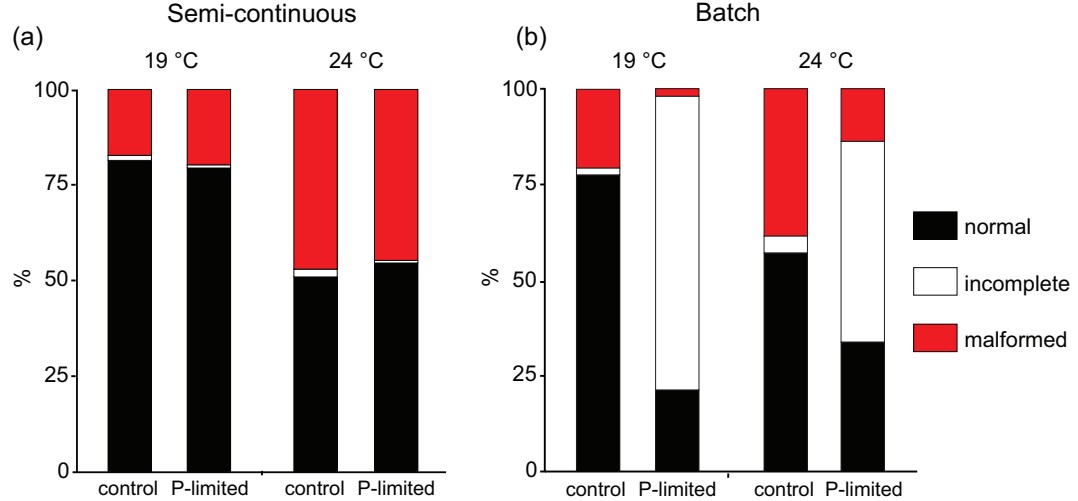

**Figure 3: Coccolith morphology of *Emiliania huxleyi* grown at 19 and 24 °C in control and P-limited medium in semi-continuous and batch culture. Coccoliths were classified into the categories normal, incomplete, and malformed; see Table 3, Fig. 2.**

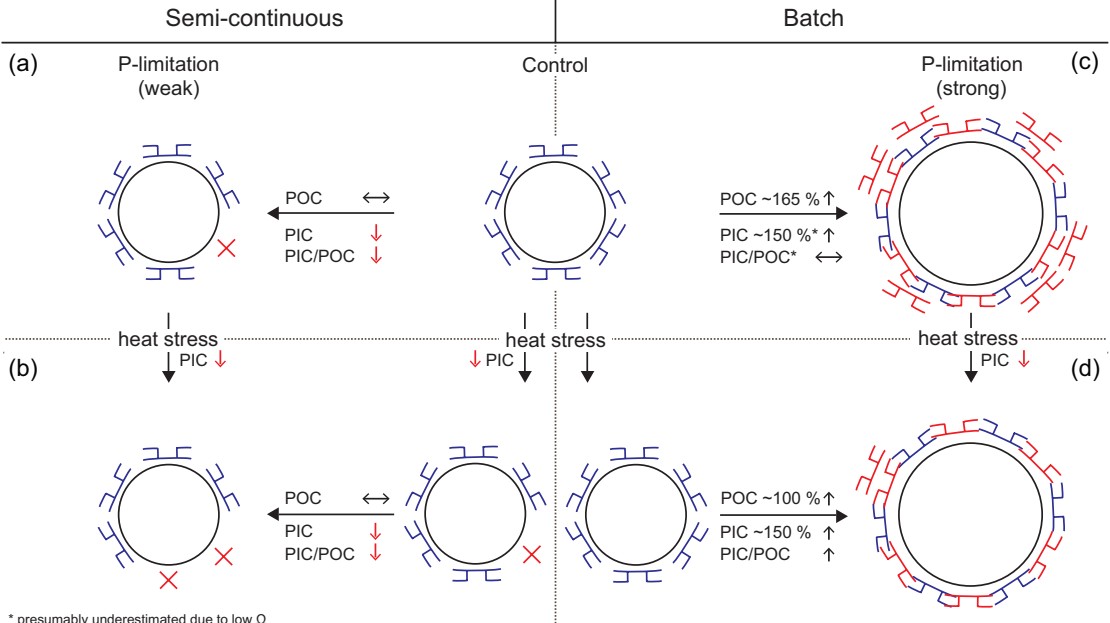

**Figure 4: Schematic of the combined effect of P-limitation and heat stress in semi-continuous (a,b) and batch culture (c,d) of *Emiliania huxleyi*. Blue coccoliths represent coccoliths covering cells of control cultures, whereas red coccoliths/crosses denote new/missing coccoliths. The asterisk (*) indicates cultures that were strongly undersaturated in calcite.**