# Peer review of "Phosphorus limitation and heat stress decrease calcification in *Emiliania huxleyi"

_Biogeosciences, 2017_

## Referee Comment (RC1) · Anonymous Referee #1 · 9 Jun 2017

The manuscript 'Phosphorus limitation and heat stress decrease calcification in Emiliania huxleyi' presents a new and valuable dataset that assesses the combined effects of two possible future environmental stressors on the coccolithophore Emiliania huxleyi. The study presents results of semi-continuous and batch culture experiments in replete (control; 10 $\mu$M) and P-limiting conditions (0.5 $\mu$M) at two temperatures (19°C and 24°C), performed with a clone of E. huxleyi isolated from the Oslo fjord.

Overall, this paper is well-structured and clear, and the rationale is justified. The new data presented are relevant, and add to a growing collection of experimental results that assess response of different coccolithophore species and strains to changing environmental conditions. The tables and figures are clear, detailed and appropriate. In general, the discussion is well-reasoned, with suitable reference to published and

available studies. The authors successfully integrate their new data into the ongoing debate, and present this in a way that is accessible to readers across disciplines. There are, however, some areas that would benefit from additional clarification:

1) The environmental significance of the data is presented with appropriate caution in the main text, but becomes over-stated in the abstract and conclusions sections.

The main interpretations of the data presented are:

- A future rise in global temperature, accompanied by a decrease in nutrient availability may decrease $CO_2$ sequestration by coccolithophores through lower overall carbon production

- The export of carbon may be diminished by a decrease in calcification and a weaker coccolith ballasting effect.

In general, the justification for these statements within the discussion section is well-balanced and makes appropriate reference to the contrasting results obtained from different species/strains and experimental setups (e.g., page 7, lines 19-31), which builds on points made in the introduction. However, these intricacies are not apparent in the concluding statements of the abstract and conclusions, which refer to 'coccolithophores' (e.g., page 1, line 14) and 'E. huxleyi' (page 9, line 33), without acknowledgement of potential species and strain-specific responses. These statements seem to require additional justification. The authors do exercise appropriate caution with their interpretations (e.g., using 'may' and 'based on this study'), but some additional context might strengthen their interpretations. For example, it would be beneficial to assess how widespread/dominant this strain of E. huxleyi is in comparison to those strains/species cited from other publications. Further recognition of the potential for acclimation is also important.

2) Assessing coccolith morphology

The criteria for classifying coccolith morphology as 'normal, incomplete, and malformed' should be included in Section 2.4 (Methods). At present, the significance of 'incomplete' coccoliths as those that have undergone secondary dissolution (rather than being 'incomplete' due to incomplete primary formation) is only discussed in Section 4.2. This is an important distinction, particularly for fellow scientists who attempt to apply the same morphologic criteria in other experiments. An additional image of a representative coccosphere from the cultures that had higher levels of malformation would also be a useful addition to Fig. 1.

3) Semi-continuous and batch culture experiments

The paper successfully highlights the reasons for performing both semi-continuous and batch culture experiments (i.e., production cannot be determined from the batch culture approach). It also describes the differences between the approaches with respect to real environmental scenarios (page 2, lines 3-7). In this regard, does the experimental approach have any relevance for the strain of E. huxleyi used? I.e., would the approach that most closely replicates it's original natural environment in the Oslo fjord be more representative for this strain?

Other minor comments:

- The uncertainty for the number of coccoliths per cell is much higher for the P-limited batch cultures at both temperatures (+/- 20 and 15). Is there a reason for this?

- Table 2 (page 5 line 16) is referred to before Table 1 (page 5, line 38)

- The significance of the red and blue colors in Figure 2 is missing

- There is no reference to the error bars in Figure 4.

---

## Referee Comment (RC2) · M. Hermoso (Referee) · 14 Jun 2017

The culture study by Andrea Gerecht and collaborators addresses the fate of coccolithophore algae in our Anthropocene ocean. The Authors did not directly examine the effect of increased ocean carbonation and decrease oceanic pH, but rather their laboratory culture study tackles more indirect, yet important, ancillary environmental change by seeking to determine how calcification by this biological group will evolve with rise in temperature and nutrient limitation in the oceans. Their experiments are based on one single strain of *Emiliania huxleyi* originating from a fjord in Norway. The authors also compare the results obtained from batch *vs* semi-continuous cultures and highlight significant discrepancies between the two techniques, which is of potential interest from a more methodological perspective. The take-home message

claimed by the team is that limiting phosphorus availability and increasing temperature (the latter being introduced as a stressor) will likely reduce carbon fixation by *Emiliania huxleyi* and diminish mineral-ballasting export of organic carbon to the deep ocean.

I am generally supportive of publication of this work in Biogeosciences. I have, however, a number of comments and questions, which I hope the Authors will find useful to prepare their revisions.

General comments

(1) More information is needed on the cultured strain of *Emiliania huxleyi*, including (where possible) the date of isolation, the morphotype of coccoliths, whether the strain is deposited in a Culture Collection (or in the process to be), and the conditions under which the stock culture is maintained in the laboratory (temperature, light irradiance, *etc*), is the strain axenic?

(2) There are no details given of the culture technique *per se* apart from strategy (batch *vs* semi-continuous) adopted. Were the cells acclimated to the target phosphorus concs and temperature conditions when proper experiments began? This is all the more important as changing temperature is conceived a stressing factor in the study. We know that coccolithophores, and singularly *E. huxleyi* is fast adapting to changing environments so this methodological aspect is crucial for the implications of culture study to wild specimens and at timescales compatible with adaptation in the natural environment.

(3) I would be valuable to elaborate on the malformations of the coccospheres/coccoliths observed by the Authors. The rationale behind the discrete

class of malformation features and the implications for biomineralisation are elusive in the manuscript. On this note the argument that 24 °C represents a heat stress for this strain as it was isolated from waters measured at 21 °C or lower, and that more malformations were observed (p. 8 lines 23-25) does not appear as a strong argument to me. Likewise, it is not entirely clear to me how the Authors are able to distinguish between malformation and dissolution features.

(4) I feel that at places the discussion is too descriptive and lacks a better attempt to understand the cellular mechanisms at play for the environmentally-driven change in carbon fixation. An integration of P acquisition strategy by *E. huxleyi* (a species with the ability to excrete ligands to increase P supply to the cell) with growth dynamics and organic and inorganic carbon fixation for each condition would be extremely useful and add value to the paper.

Specific comments

- Page 3 Lines 6-7: In my opinion, looking at Table 3 it is not nutrient limitation that limits further algal growth in thus set-up, but rather the drift in the carbonate chemistry of the medium (see e.g. Hermoso 2014 in Cryptogamie, Algologie 35(4):323-351). More broadly, I do not believe that the stationary phase represents an end-of-bloom scenario.

- Page 3 Line 19: inter alia?

- Page 3 Line 22: Carbon "fixation" rather than "production"

- Page 3 Lines 22-24: There a many other references (Bollmann et al. 2010 in Protist

161:78–90; McClelland et al. 2016 in SciReport 6:34263 *etc etc* of which some cited at the end of the discussion should be also mentioned here).

- Page 3 Line 30: I still think that "heat stress" is not appropriate here for the reasons outlined in general comments. The effect of changing temperature from 19 to 24 °C on growth rate is not very detrimental (by 10 percent) compared to the effect of other manipulations of the culture medium in literature. *E. huxleyi* has a broad tolerance and adaptability to temperature change compared other taxa, such as *C. pelagicus*.

- Page 6 Line 26-27: I disagree with this statement. Also Table 3 should be given the starting conditions.

- Page 7 Line 1-2: How about the number of layers of coccoliths forming the spheres? This could be useful to put in the context of the dynamics of cell division.

- Page 7 Lines 35-36: The Authors should add a discussion on the mechanisms for this observation. There are a few studies on cells being stuck in the haploid phase due to the lack of N and P provision to replicate DNA and allow further division.

- Page 8 Lines 8-9: Please refer to recent study on this particular point by Aloisi in Biogeosciences 15: 4665-4692, and incorporate suitable discussion on the mechanisms.

- Page 9 Lines 6-7: I do not follow the argument being made here. Please clarify.

- Page 10 Lines 1-2: I recommend that the Authors tone this down, as we know that such a conclusion at the scale of the global biogeochemical cycle requires longer-term

and multi-strain investigation although I appreciate the "may" being used here.

Sincerely,

Michael Hermoso

---

## Referee Comment (RC3) · Anonymous Referee #3 · 21 Jun 2017

Review of Gerecht et al. for Biogeosciences (2017)

In general the article is a worthwhile contribution. They find that P-limitation can actually decrease the PIC quota, PIC production, and PIC/POC ratios in E. huxleyi, which is opposite that which has been most commonly reported in earlier literature, although some more recent studies are cited to report similar results. This contrast is little discussed. Neither the P-stress nor the heat stress used is very clearly justified. Where are such changes predicted to occur? Why is P-stress chosen instead of N-stress, when much more of the world's ocean is thought to show N-limitation of primary production? It is especially not clear to me what natural conditions are mimicked in the P-limited batch cultures. Do E. huxleyi blooms naturally experience these chemical conditions (e.g., such low DIC and omega-calcite values)? If these are conditions that

only arise in batch cultures at very high cell densities only reachable in lab monocultures, perhaps they must be more careful of extrapolating their results from stationary phase cultures to changes in carbon export.

In terms of heat stress, it's not made very clear why the temperatures of 19ËŽ and 24ËŽ were chosen, although there is some justification given in the Discussion. Is 19ËŽ a typical SST in the North Sea (assuming the clone here represents a North Sea population) or typical of the Oslo Fjord? With global warming is it expected to reach 24ËŽ regularly? What about E. huxleyi populations currently found in 8-12ËŽ waters, would the same tendencies occur if grown at from 13ËŽ to 19ËŽ?

It's not necessary for all studies to try to replicate specific environmental conditions (often impossible), perhaps especially when the goal is to understand physiological limits or to take a first approximation. However, considering this lack of grounding of experimental design within an explicit environmental context, it appears that the Conclusions should be more cautious in extrapolating to biogeochemical effects.

There is a focus on biogeochemical effects, but nothing on ecological effects of the documented changes in PIC and coccoliths. What function do they serve? I might suggest the review by Monteiro et al. to look at, and think of some of the consequences.

Finally, I'm not so sure of the extent of the novelty of this study. They say "To our knowledge, this study is the first to specifically test the impact of heat stress...", but then there actually are a few quite relevant studies (it depends on how "heat stress" is defined), some of which they cite. For that reason, a more rigorous study design in an explicit environmental context would have been much stronger.

Considering the careful criticism, I think it will be worth accepting the paper once the issues I have identified have been addressed. I do not foresee that the authors will have trouble resolving these issues.

I also have several issues, discussed point-by-point below, about the details (or the details they provided) about experimental design, analyses, and some of their introduction or discussion of the relevant literature. These must be resolved.

p. 1 Line 26, should probably cite something more recent as well, such as the meta-analysis by Meyer and Riebesell 2015.

I don't understand the justification of P-stress, as opposed to perhaps N-stress, considering that much more of the ocean is thought to be N-limited than P-limited.

p. 2 Lines 6-9 : "Batch culture on the other hand represents an end-of-bloom scenario in which the lack of nutrients limits further cell division... production cannot be determined in the batch approach ". That's only true if the last part of a batch culture is analyzed, as growth becomes limited due to exhaustion of nutrients, build-up of metabolites, shading, limited gas exchange, etc. In fact, there are many published experiments where production rates were determined in dilute batch culture, in the early exponential phase of growth before DIC consumption or nutrient consumption was substantial. Line 29: "None of these studies, however, tested the effect of above-optimum temperature ". I don't understand this unless one defines what is "above-optimum temperature". In the study presented here the temperatures used are 19ËŽ C and 24ËŽ C. The study of Feng et al. (2008) (which they cite) used 20ËŽ C and 24ËŽ C, so if 24ËŽ C is an above-optimum temperature in the present study, why wasn't it in Feng et al.? Rosas-Navarro et al. 2016), which they also cite, used 25ËŽ as the highest temperature. How is "above-optimum temperature" defined, and wouldn't it possibly depend on the origin of the cultures? For example, E. huxleyi from lat. 50ËŽ in the North Atlantic probably does not experience such temperature, E. huxleyi in the Mediterranean Sea will rarely experience temperatures that high, and those may be unexceptional surface temperatures for the tropical Pacific, where E. huxleyi can also be found. This is discussed much later on p. 8, lines 20-26, but I am not so convinced how relevant this temperature range is. Lines 36-37: Should consider (and cite) also work of van Bleiswijk et al. 1994 and Rokitta et al. 2016 very relevant for theme of E. huxleyi response to P-limitation. p. 3 I have a problem with the use of K medium for nutrient

experiments. K medium contains a mix of ammonium and nitrate as N-source, and it contains glycerophosphate as a P source. It's not clear from Gerecht et al. (2014) if they modified these components. They need to give the basal medium composition they used. What volume were cultures? p.4 For semi-continuous cultures, what was the dilution rate or growth rate? How was it determined or confirmed that the cultures in fact were limited by P-limitation in the semi-continuous cultures? How could maximum cell concentrations have reached 170000 cells/ml if cultures were diluted back to 10000 cells/ml every second day? To have reached 170000 cells/ml from 10000 cells/ml in only 2 days would require aprox. 4 cell divisions per day, which has never been reported for this species. Is this due to experimental or measurement errors? Line 7: "P-limited cultures were harvested in stationary phase, . . ." for how long in stationary phase? This is not clear from Fig. 4. p. 5 Line 10: Give manufacturer & city for "CASY" Line 30: "Average values were compared by a t-test". Was this pairwise test performed after the two-way ANOVA? If so, with what correction for multiple comparison? They are testing two factors (T and P-limitation) so should be doing a two-way ANOVA, not t-tests. p. 6 Line 24 and Table 3: What limited the growth of control batch cultures?

p. 8 Lines 10-12: "These large "ready-to-divide" cells (Gibbs et al., 2013) not only accumulate POC, but also accumulate PIC, leading to the 2-3-fold increase in coccolith number per cell observed in stationary phase cultures (Fig. 2c,d)." How do you know these cells are "ready-to-divide"? If they really are "ready-to-divide", do you mean they are blocked in G2 or M phase of the cell cycle? That doesn't make much sense.

Lines 10-12: This is an important justification for their selection of temperatures. Nevertheless, I'm not very convinced about how these temperatures aare reoe. I would prefer them to explicitly give the range of temperatures experienced in the North Sea as well as the fjord. Why is 19ËŽ C a "normal temperature"? What does that mean?

Lines 12-13: "Stationary phase can be likened to an end-of-bloom scenario in nature, during which E. huxleyi sheds numerous coccoliths, leading to the characteristic milky color of coccolithophore blooms". Maybe, but it's also well know that the end of E.

huxleyi blooms involves infection by the virus EHV.

p. 9 Line 10 "The percentage of partially dissolved coccoliths was higher at normal temperature than under heat" Where is this shown? Data is presented on "incomplete", "malformed", and "normal" coccoliths in Fig. 3. They state "The high numbers of incomplete coccoliths observed in P-limited batch cultures were likely a result of secondary dissolution (Fig. 1d; Langer et al., 2007) due to the low calcite saturation state reached in stationary phase cultures." I would like to see more examples of incomplete coccoliths. Perhaps they can show that the type of incompleteness that appears in P-limited batch cultures (when omega-calcite is less than 1) is distinct from what appears when omega-calcite is greater than one?

Overall responses: 1. Does the paper address relevant scientific questions within the scope of BG? Yes. 2. Does the paper present novel concepts, ideas, tools, or data? Some novelty. 3. Are substantial conclusions reached? Somewhat. 4. Are the scientific methods and assumptions valid and clearly outlined? I have outlined some places where further details or considerations are needed. 5. Are the results sufficient to support the interpretations and conclusions? Mostly. 6. Is the description of experiments and calculations sufficiently complete and precise to allow their reproduction by fellow scientists (traceability of results)? I have detailed several points that need to be corrected, but I anticipate the authors will have no problem responding. 7. Do the authors give proper credit to related work and clearly indicate their own new/original contribution? Mostly. I indicate a few places where they might consider or cite other works. 8. Does the title clearly reflect the contents of the paper? Yes. 9. Does the abstract provide a concise and complete summary? Yes. 10. Is the overall presentation well structured and clear? Yes. 11. Is the language fluent and precise? Yes. 12. Are mathematical formulae, symbols, abbreviations, and units correctly defined and used? Yes. 13. Should any parts of the paper (text, formulae, figures, tables) be clarified, reduced, combined, or eliminated? I mention several points that need clarification, but I do not specify exactly how they can do this. They could, for instance, use more images of

coccoliths in different states to show how they have classified. 14. Are the number and quality of references appropriate? Yes, but I suggest a couple more. 15. Is the amount and quality of supplementary material appropriate?

---

## Referee Comment (RC4) · Anonymous Referee #4 · 4 Jul 2017

General Comments:

Gerecht and co-authors present an interesting study that investigates a two-stressor 'future ocean' scenario of phosphorus limitation combined with beyond thermal optimum temperatures. Concurrently manipulating two primary controls on physiology is a much need step beyond understanding how variability in single conditions affect phytoplankton and is of suitable scope for publication in Biogeosciences. However, I feel that the discussion is weakened by an emphasis on comparing two culturing methods rather than comparing individual vs. interactive effects of the stressors in question. The specific choice of experimental conditions is also poorly justified and not placed into context. I feel that once the points raised by myself and the other reviewers are addressed, this manuscript will be suitable for publication in Biogeosciences.

[Figure]

Specific Comments:

1. In the broader interpretations of their calcification results, the authors state in the Abstract, Introduction and Conclusions that decreases in calcification rates in E. huxleyi could "lessen the ballasting effect of coccoliths and weaken carbon export out of the photic zone", or similar wording. In support, they reference Ziveri et al. (2007), who conclude that, despite the high abundance of Emiliania relative to other coccolithophore species, the small size and very low species-specific carbonate mass of their coccoliths means that they consequently export far less carbonate than expected. Baumann et al. (2004) similarly concluded that Emiliania plays only a relatively minor role in carbonate export in the Equatorial and South Atlantic. Can the authors support their statements of alterations to the carbon cycle more quantitatively using their PIC production values and abundances of Emiliania in the field? Or provide references of studies that better support these statements?

2. Considerable emphasis is made on the short-comings of the batch culturing technique compared to the semi-continuous culture technique. Comparison of these two methods is present throughout the results and discussion, and, in my opinion, obscures a clear and explicit evaluation of contrasting individual (warming only, P-limitation only) vs. interactive (warming and P-limitation) effects and evidence (or not) for positive interactions. It is stated that batch culture experiments can only represent a severely nutrient depleted scenario whereas a semi-continuous set-up provides an acclimated low-nutrient population (p3, ln 7; p7 lns 12-17). I think that these statements are somewhat misleading for the following reason: A batch culture experiment experiences exponential growth at whatever the starting nutrient concentrations until these nutrients become sufficiently depleted that exponential rates of growth can no longer be maintained and growth rate rapidly falls to zero. A semi-continuous culture is just a batch culture that is subcultured/diluted (typically) around mid-exponential-phase cell concentrations several times. I therefore find it strange that the authors did not just sample their P-limited batch culture experiment at mid-exponential phase (as they did with the

control experiments) well before the 'severe' nutrient depletion of stationary phase began, which based on Fig. 4 would have meant sampling on day 3 or 4. Would this not have been a more realistic comparison of control and P-limited conditions during exponential phase in all experiments? It would also allow a comparison of multiple generations of growth (semi-continuous) with far fewer generations of growth (batch to mid-exponential phase). It would also have removed the issues with calculating production rates, as only exponential growth phase populations would have been sampled.

3. I would like to see a broader context of the area where a similar degree of temperature and phosphorus stress is predicted to be experienced in the context of this strain isolated from a Norwegian fjord. Similarly, there is no discussion of the fact that physiological stress experience by one strain of this species under climate change is as likely to lead to its ecological replacement by another, more tolerant strain given recent studies presenting the large genetic pool of Emiliania (e.g., Read et al., 2013, which also discusses differences in genes for tolerance of low phosphorus conditions between strains).

4. I was surprised to see that no figures of any POC, PIC, or POP data were presented, only data in the tables. Was there a reason for this? Given the two experimental approaches, two temperatures and two nutrient states, it made it difficult to quickly visualise the dataset.

5. The strain used seems to be a new isolate – do the authors intend to deposit this strain into a culture collection for use by other researchers? Given it is not held in a culture collection, the authors must provide the essential ancillary information on the isolate and its maintenance in culture.

6. It is stated deep into the discussion that the temperature at the isolation location does not exceed 21 degrees and this is presumably how the authors decided that a temperature of 24 degrees was beyond the thermal optimum. Did the authors perform a systematic temperature optimum assessment by determining growth rates at a range

of temperatures? Given that the exponential growth rates were not substantially different between temperature treatments (semi-continuous) and in fact were higher in the 24 degree treatment for the batch culture approach, this would perhaps suggest that (using exponential growth rate as a physiological indicator) this isolate has a relatively broad thermal tolerance.

7. The authors do not state whether there was any period of acclimation for populations experiencing low phosphate or high temperature treatments.

8. How did the authors account for the tendency of Emiliania to form multi-layer coccospheres when counting the number of coccoliths from SEM images? Comparing the coccosphere size from CASY with the cell size from light microscopy (back calculated from the volume data) and considering the thickness of Emiliania coccoliths, would suggest that coccospheres were not mono-layer.

9. Is there any reason why the authors refer to both light microscope and CASY cell size measurements as "cell size" when CASY gives coccosphere size measurements? There are huge differences in volumes between the two methods due to the coccosphere and whilst cell size directly relates to cell carbon, coccosphere size does not and therefore this unnecessarily confuses the reading in parts.

10. The discussion would benefit from a discussion of the physiological mechanisms behind the observed response to P-limitation and heat stress singularly and combined, and there is considerable literature on this species, other coccolithophore species and other phytoplankton groups that would support such a discussion.

p6, ln 37-39 – referred to changes in size but "twice as large in stationary phase" refers to cell volume, so this should be changed to reflect this.

p8, ln 7-9 – The reference to Sheward et al. (2016) on line 7 should be changed to Gibbs et al. (2013) as the latter presented Emiliania data. This sentence could also include C. braarudii, Calcidiscus and Helicosphaera from Sheward et al. (2017). The

Sheward et al. (2016) at the end of line 9 should be changed to 2017 (I think you have referred to the discussion paper rather than the finally-published article).

Technical Corrections:

Throughout the paper, there are inconsistencies with the author order in your references. Sometimes they are ordered by date, other times alphabetically, and several times I can find no logic to the order! (e.g., p2, ln 37-38).

---

## Referee Comment (RC5) · Anonymous Referee #5 · 7 Jul 2017

General comments

The manuscript -Phosphorus limitation and heat stress decrease calcification in Emiliania huxleyi - by Gerecht et al. addresses and try to answer to an essential question in the coccolithophores community : How coccolithophores physiology will change in a environmental conditions changing ocean ? With this wider question in mind, this work focused on a more specific question : How Emiliania huxleyi will react to futur nutrient (P) and temperature conditions in ocean in term of calcite production, nutrient (P) requirement and morphology?  The two different experimental methods used allows the comparison between the response of a strain of Emiliania huxleyi cultivated in a closed (batch) system and in semi-continuous cultures. This latter allow a determination of calcite and POC production and a lower nutrient limitation in contrary of a batch

approach for which the calculation of production is impossible due to a decrease of growth rate and which led to a strong limitation of cells. The conclusions of this paper are mainly the negative feedback on carbon sequestration by the decreasing of cell density of Emiliania huxleyi in a warmer ocean due to a higher P-requirement under heat stress.

These both experimental approaches are a good way to have indication on the method dependence for a given response. Therefore, this work brings new insights on the response of Emiliania huxleyi to a P-limitation combined with a heat stress as well as new data to the literature known about Emiliania huxleyi physiology for both batch and semi continuous approach and for both P-limitation and temperature stress conditions. However, the choice of the cultures conditions, P limitation and temperature stress, need to be express with more details in the introduction or in the methodology part concerning the choice of the values. The paper is well written and has a clear and easy-to-read structure. This manuscript will fit perfectly to the journal Biogeosciences after few corrections and comments from reviewers. The figures and tables are well presented and show essential results that you discuss in this manuscript.

Specific comments

p. 2 ; Ln 9 : You need to add the word particulate to present for the first time in your manuscript the PIC and POC terms.

p. 3 ; 19 : What does inter alia mean?

p.3 ; Ln 29 & 35 : Did you test different temperature on the strain (before these experiment) to find the optimal temperature at the selected light dose (12 :12, 100$\mu$mol photons.m-$^2$.s-1) ? Or did you have an idea (or reference) of the optimal temperature of your strain before selecting your two temperature conditions ?

p.3 ; Ln 34-39 : What is the name of your strain ? Have you done some acclimations of your strain to the temperature and P concentration conditions before starting your

experiment ? You should indicate in your method if you did. What is the initial nitrate concentration in your medium ? Is it the standard concentration of NO3 in a K medium ? If not, the value of the concentration need to be indicate in this method part. You also need to add the value of the salinity of your medium.

p. 5 ; Ln 21-26 : Did you take into account that Emiliania huxleyi can have multiple layers of coccoliths ? It is clearly visible on your figure 1 that your strain can have several layers of coccoliths depending of the cultures conditions. How did you take this particularity in your coccoliths counts ? What is the standard deviation on your counts ? At least, you have triplicate so you need to specify the standard deviation on your number. What about coccospheres diameter on your SEM images ? Did you think about doing coccopheres measurements ? If not, is it because of the high number of detached coccoliths on your filter ? It should be interesting to discuss the PIC content related to your coccolith number and the thickness of your coccolith layer.

p. 6 ; Ln 24-34 : Final cell density for batch cultures were being push really high in order to get the P-limitation. However, respect to LaRoche et al. 2010, inorganic carbon system changes need to be kept below 5% to avoid carbon system changing due to the high cell density. This meens that nutrient limitation experiment for batch system have to attain a reasonable final cell density in order to keep a DIC system quasi-constant through the experiment and to avoid a inorganic carbon limitation before the P limitation in this case. In your P-limited cultures, DIC changes that you get at the end of the experiment led to a low calcite saturation constant. Therefore observations in morphology and calcite content may be due to this changes rather than the P-limitation and heat stress effect. You took into account this changes in your discussion but it will be useful to indicate why you did batch experiments in this way. Could you justify this choice in the methodology of your batch experiment ? Did you think about carrying the experiment with a lower initial P concentration rather than push the cell density so high ? Or did you have a target initial P concentration that you wanted to test ? In this last case, a comment on the 2.1 part will be necessary.

p. 7 ; Ln 27 : What is the normal temperature? Even if you explain it later (p.8 Ln 26), you should describe here that the normal temperature is 19°C if you want to use this term. It is not obvious for readers.

p. 8 ; Ln 1-5 : same comments that previously on the DIC system

Figures & Tables

Figure 2 : You need to add the color explanations : blue is your schematic initial cell, red is the schematic effect observed at the end of your experiment.

Figure 4 : You should add the error bar on triplicate somewhere in your legend.

Tables 1, 2 & 3 : You should add the standard deviation in your legend and your n number (for example, n = 3 if it is triplicate).

Table 2 : You need to clarify in the legend of your table if the cell volume has been calculated with measurements of cell diameter from the harvest day or from an average of daily measurements.

———————————————————

---

## Author Comment (AC1) · 11 Aug 2017

Authors' reply to reviewers' comments:

We would like to thank all five reviewers for their positive and supportive evaluation of the manuscript and for their suggestions on improvements. We have carefully considered all comments and hope to have addressed each comment in a satisfactory manner. Here, we first address some common points that were raised by most/several reviewers, before responding to the reviewer's general comments, followed by the specific comments to the text.

Common points raised:

1) Several reviewers requested more information on the particular strain used in the

current study (referees #2, #4, #5).

Authors' response: Additional information on the origin and isolation of this strain has been added to the methods section, including date of isolation and the conditions under which the stock culture is maintained in the laboratory. The strain is not axenic and has been deposited at the NIVA culture collection (niva-cca.com). This strain belongs to morphotype A, which is widespread in the Northeast Atlantic.

2) Several reviewers required further details on how the coccolith morphology was assessed (referees #1, #2), especially in regard to distinguishing between "incomplete" and "dissolved" coccoliths (referees #1, #2, #3).

Authors' response: We have defined three categories of coccolith morphology: normal, malformed and incomplete. We have added a table to the methods section to outline the characteristics of these categories. We do not differentiate between different kinds of malformations, only between incomplete (considered to be of normal morphology, but are uniformly unfinished) and malformed (of overall irregular morphology or with irregular morphological features). Calcite content of single malformed coccoliths can be expected to be lower, the same or higher than normal coccoliths depending on the kind of malformations, whereas incomplete coccoliths can be expected to contain less calcite (though we did not measure this). We could clearly distinguish between incomplete and malformed coccoliths on the SEM images.

Our data set is complicated by the occurrence of high amounts of dissolved coccoliths in P-limited batch cultures, the features of which we describe in the new table. This dissolution affects all categories of coccoliths (normal, malformed, incomplete). However, the features of dissolved coccoliths are similar to those of incomplete coccoliths and we were not able to make an unambiguous distinction between the two possible origins of "incomplete coccolith morphology" i.e. "incompletely produced" or "incomplete because of secondary dissolution". We therefore only have one category of "incomplete coccoliths" in our Fig. 3, most of which are a result of dissolution in P-limited

stationary phase batch. We have added an upfront description of these differences in the methods section as well as additional images to Fig. 1.

3) Referee #2 questioned the validity of using 24°C as supraoptimal temperature and most reviewers required further justification for the choice of the specific culture conditions (referees #2, #3, #4, #5).

Authors' response: The main evidence that these cultures were heat-stressed is provided by the decrease in growth rate at 24°C compared to 19°C. The decrease is slight (6-9%, Table 1), and only observed in semi-continuous, not batch cultures. However, measurements of growth rate in semi-continuous cultures are more robust because growth rate is measured as an average of numerous dilution cycles. A decrease in growth rate at a higher temperature is the definition of above-optimum growth. According to Eppley (1972), phytoplankton growth rates increase with increasing temperature as long as this temperature is below the optimum for growth, whereas above optimum temperature, growth rates decline. As growth rates decline more sharply at above-optimum temperatures (i.e. heat stress is more detrimental than below-optimum temperature), it is often difficult to culture phytoplankton at above-optimum temperature. Therefore, obtaining a stronger difference in growth rate at even higher temperature would have been technically difficult and would likely have resulted in the crash of the culture. The isolation temperature and increased presence of malformations at 24°C are additional indications that this temperature was above optimum. These motivations and definitions have now been mentioned upfront in the methods section.

We agree with referee #5 that it would have been beneficial to rigorously test growth rates over a broad range of temperatures before choosing the applied temperatures of 19 and 24°C. The fact that we did not do this, however, does not detract from the fact that 24°C was above the optimum for growth.

4) Several reviewers expressed concern about possible DIC-limitation in P-limited batch cultures (referees #2, #3, #5).

Authors' response: As outlined in the text, we discuss that P-limited batch cultures may have been co-limited by P and DIC at the time of harvest. However, entry into stationary phase was due to P-limitation for the following reasons: There was a drastic decrease (3.3-3.7-fold; Table 1) in the POP content of "P-limited stationary phase cultures", which is a sign of strong P-limitation and would not be expected under DIC limitation. Secondly, POC fixation continued in stationary phase leading to a strong increase in cell size/POC content, which again argues against DIC limitation. The recent publication by Wördenweber et al. (2017) gives mechanistic support for this observation. By analyzing the metabolome, these authors observed that under P-starvation metabolites such as lipids are accumulated i.e. enzymatic functionality is preserved, whereas enzymatic functionality ceases under nitrogen starvation. This continuing fixation of POC by non-dividing cells led to the high consumption of DIC in P-limited batch cultures. Thirdly, Buitenhuis et al. (1999) calculated a minimum DIC-threshold for calcification in E. huxleyi of ca. 600 $\mu$M. So our DIC-values were arguably only at the threshold for DIC –limitation.

5) Several reviewers questioned the validity of using stationary phase as an "end-of-bloom" scenario (referees #2, #3).

Authors' response: It is difficult to recreate natural situations in the laboratory and we do not pretend to have faithfully recreated an "end-of-bloom" scenario as bloom demise is likely regulated by many factors, such as viral lysis. However, "stationary phase batch cultures" are the closest approximation we could achieve in the laboratory to an "end-of-bloom" scenario and it would now be interesting to test this hypothesis in the field.

Anonymous Referee #1:

1) The environmental significance of the data is presented with appropriate caution in the main text, but becomes over-stated in the abstract and conclusions sections.

Authors' response: We agree that general statements on the effects of P-limitation and

heat stress on E. huxleyi or coccolithophores as a whole, require further investigation of additional strains and/or species to account for strain- and species-specific variability. We have therefore modified the relevant section of the abstract (coccolithophores now reads E. huxleyi) and conclusions (E. huxleyi now reads "a temperate strain of E. huxleyi") to specify that our interpretations are specific to E. huxleyi/the strain under study. We have also elaborated on which physiological effects may be more general and which require further studies as outlined below.

The main interpretations of the data presented are:

- A future rise in global temperature, accompanied by a decrease in nutrient availability may decrease $CO_2$ sequestration by coccolithophores through lower overall carbon production.

Authors' response: As an additional justification of the above hypothesis, we would like to point out the following. An increase in phosphorus requirements at higher temperature has been observed also for Coccolithus pelagicus (Gerecht et al., 2014) and can be inferred from the data presented in Satoh et al. (2009) and Feng et al. (2008) for two additional strains of E. huxleyi, as discussed on p. 8, l. 30-33. As in the present study, in Gerecht et al. (2014), higher P-requirements led to lower final cell concentrations in P-limited stationary phase cultures of C. pelagicus i.e. lower biomass under P-limitation. Increased phosphorus requirements leading to lower overall carbon production in a future warmer ocean may therefore be a general feature of coccolithophores. We have slightly rewritten the relevant section of the discussion and conclusions to better highlight this point.

- The export of carbon may be diminished by a decrease in calcification and a weaker coccolith ballasting effect.

Authors' response: A decrease in coccolith coverage at high temperature has also been observed for C. pelagicus (Gerecht et al., 2014) as stated on p. 9, l. 8-9. However, this is to our knowledge the first study to describe a decrease in calcification under

weak (i.e. not affecting growth rate) P-limitation. On the contrary, in C. pelagicus weak P-limitation decreases POC over PIC production (Gerecht et al., 2015). So further studies on other strains and species are necessary to confirm how widespread this physiological response is. We have stated this accordingly in the conclusions.

For example, it would be beneficial to assess how widespread/dominant this strain of E. huxleyi is in comparison to those strains/species cited from other publications.

Authors' response: Although we have no information on the dominance of this particular strain, it belongs to morphotype A that is widespread in the Northeast Atlantic, while morphotype B dominates in the North Sea (van Bleijswijk et al., 1991). However, considering the physiological variability within a morphotype, e.g. Langer et al. (2009), we cannot make any conclusions about the general physiological responses based only on information of the morphotype.

Further recognition of the potential for acclimation is also important.

Authors' response: Cultures were acclimated to experimental conditions (temperature, P-concentrations) for ca. 10 generations before starting the experiment (this has been added to the methods section). Ten generations should be sufficient for acclimation to experimental conditions, according to the model of Aloisi (2015). Longer-term adaptation to changing environment is a factor that needs to be considered and we have included a short discussion about the potential for adaptation in the discussion.

2) Assessing coccolith morphology

The criteria for classifying coccolith morphology as 'normal, incomplete, and malformed' should be included in Section 2.4 (Methods). At present, the significance of 'incomplete' coccoliths as those that have undergone secondary dissolution (rather than being 'incomplete' due to incomplete primary formation) is only discussed in Section 4.2. This is an important distinction, particularly for fellow scientists who attempt to apply the same morphologic criteria in other experiments.

[Figure]

Authors' response: Please see common point (2).

An additional image of a representative coccosphere from the cultures that had higher levels of malformation would also be a useful addition to Fig. 1.

Authors' response: We have added additional images to illustrate the different classes of coccoliths in more detail in Fig. 1, and have included a representative coccosphere with high levels of malformed coccoliths.

3) Semi-continuous and batch culture experiments... In this regard, does the experimental approach have any relevance for the strain of E. huxleyi used? I.e., would the approach that most closely replicates its original natural environment in the Oslo fjord be more representative for this strain?

Authors' response: Emiliania huxleyi blooms can occur in the Oslo fjord, usually after the diatom spring bloom. So a bloom scenario is certainly relevant for carbon production by this strain in this area. However, during the rest of the year, E. huxleyi is likely present in low numbers so that also a simulated "continuous low nutrient" environment is relevant for this strain. Here, we would like to note that we wanted to test the physiological limits of this particular strain in our simulated growth scenarios (batch and semi-continuous culture). These scenarios need to be applied with caution to the more complex, natural system.

Other minor comments:

- The uncertainty for the number of coccoliths per cell is much higher for the P-limited batch cultures at both temperatures (+/- 20 and 15). Is there a reason for this?

Authors' response: P-limited batch cultures were harvested in stationary phase in which the number of coccoliths per cell had increased considerably. At the same time, the variability in coccolith coverage was higher in the stationary growth phase whereas the number of coccoliths per cell was more constrained in exponentially growing cultures. The higher uncertainty is therefore not methodological, but due to higher

biological variability in coccolith coverage in stationary phase.

- Table 2 (page 5 line 16) is referred to before Table 1 (page 5, line 38)

Authors' response: The numbering of the tables has been changed.

- The significance of the red and blue colors in Figure 2 is missing

Authors' response: This information has been added to the figure legend.

- There is no reference to the error bars in Figure 4.

Authors' response: Reference to the error bars has been added to the figure legend.

M. Hermoso, Referee #2:

(1) More information is needed on the cultured strain of Emiliania huxleyi, and the conditions under which the stock culture is maintained in the laboratory.

Authors' response: Please see common point (1).

(2) There are no details given of the culture technique per se apart from strategy (batch vs. semi-continuous) adopted. Were the cells acclimated to the target phosphorus concs and temperature conditions when proper experiments began?

Authors' response: Cultures were acclimated for ca. 10 generations to the two phosphate concentrations and temperatures before starting the experiments. This information has been added to the methods section.

This is all the more important as changing temperature is conceived a stressing factor in the study. We know that coccolithophores, and singularly E. huxleyi is fast adapting to changing environments so this methodological aspect is crucial for the implications of culture study to wild specimens and at timescales compatible with adaptation in the natural environment.

Authors' response: The E. huxleyi strain used in this study was isolated from the Oslo Fjord in the summer of 2011 and was kept in the lab as stock culture at 12°C until

running the experiments in the fall of 2013. Cultures were acclimated for about ca. 10 generations at 24°C before starting the semi-continuous cultures which ran for 10 dilution cycles with ca. 3-4 generations between dilutions so another 30-40 generations. As the growth rate was ca. 8% lower at 24°C than at 19°C we observed no indications for short-term acclimation to this higher growth temperature. This growth temperature may therefore have been at the physiological limit for this particular strain. As other strains of E. huxleyi grow at higher temperature it is likely that this strain could adapt to higher growth temperatures over longer time periods.

(3) It would be valuable to elaborate on the malformations of the coccospheres/coccoliths observed by the Authors. The rationale behind the discrete class of malformation features and the implications for biomineralisation are elusive in the manuscript. ... Likewise, it is not entirely clear to me how the Authors are able to distinguish between malformation and dissolution features.

Authors' response: Please see common point (2).

On this note the argument that 24°C represents a heat stress for this strain as it was isolated from waters measured at 21°C or lower, and that more malformations were observed (p. 8 lines 23-25) does not appear as a strong argument to me.

Authors' response: Please see common point (3).

(4) I feel that at places the discussion is too descriptive and lacks a better attempt to understand the cellular mechanisms at play for the environmentally-driven change in carbon fixation. An integration of P acquisition strategy by E. huxleyi (a species with the ability to excrete ligands to increase P supply to the cell) with growth dynamics and organic and inorganic carbon fixation for each condition would be extremely useful and add value to the paper.

Authors' response: The reviewer points out that E. huxleyi has a particularly high capacity for obtaining P from its environment, e.g. Riegman et al. (2000). This char-

acteristic makes it even more intriguing that a weak P-limitation, as imposed by our semi-continuous set-up, in which P is still readily available to the cells, should have an effect on the calcification rate. The few studies that have addressed the effect of P-limitation in a continuous setup observed either an increase in calcification (Riegman et al., 2000; Paasche, 1998; Paasche et al., 1996) or no change (Borchard et al., 2011). Please note that the increase in calcification was observed in co-occurrence with the decrease in growth rate in the continuous cultures so there is the overriding effect of growth rate. To our knowledge we present, for the first time, growth rate independent data on changes in calcification in E. huxleyi to show that calcification can decrease under P-limitation. We have some mechanistic understanding of the increase in calcification in stationary phase, mostly based on the work of Müller et al. (2008) i.e. cells are kept in the G1 (assimilation) phase of the cell cycle as P is lacking for cell division and therefore coccolith production (and to some extent also POC production) continues. However, we find it difficult to speculate about possible cellular mechanisms regarding the decrease in calcification under weak P-limitation as there is no knowledge base in the literature to explain why calcification would have specific P-requirements. On the contrary, most data point to a stronger dependence of POC than PIC production on P-resources, as exemplified by the increase in PIC/POC in stationary phase cultures. We elaborate on this in the revised discussion.

Specific comments:

(1) Page 3 Lines 6-7: In my opinion, looking at Table 3 it is not nutrient limitation that limits further algal growth in thus set-up, but rather the drift in the carbonate chemistry of the medium (see e.g. Hermoso 2014 in Cryptogamie, Algologie 35(4): 323-351).

Authors' response: If the reviewer is referring to the change in pH as "drift in the carbonate chemistry" than the change in pH was not great (lowest pH=7.7) due to the opposing effects of photosynthesis and calcification on medium pH. This pH-value is unlikely to have affected growth rates. If the reviewer is referring to the decrease of DIC as "drift" than DIC is also a nutrient that may have been limiting at the end of the

experiment. However, although P-limited batch cultures at the time of harvest were possibly co-limited by P and DIC as outlined in the text, entry into stationary phase was due to P-limitation for the reasons outlined in common point (4).

(2) More broadly, I do not believe that the stationary phase represents an end-of-bloom scenario.

Authors' response: Please see common point (5).

(3) Page 3 Line 19: inter alia?

Authors' response: This expression has been changed to "among other things".

(4) Page 3 Line 22: Carbon "fixation" rather than "production".

Authors' response: Carbon production is a technical term referring to the calculation of production based on cellular elemental content (e.g. POC) and growth rate, e.g. Langer et al. (2013). This term is widely used in the literature and is the term we are referring to here.

(5) Page 3 Lines 22-24: There are many other references (Bollmann et al. 2010 in Protist 161:78–90; McClelland et al. 2016 in SciReport 6:34263 etc etc of which some cited at the end of the discussion should be also mentioned here).

Authors' response: Although numerous studies, such as those mentioned by the reviewer, have studied the effect of temperature on carbon production and coccolith shape and size, only a handful of studies have specifically examined the effect of temperature on coccolith malformations (as defined in this manuscript). We have specified that we are referring to this aspect of temperature effects in the text.

(6) Page 3 Line 30: I still think that "heat stress" is not appropriate here for the reasons outlined in general comments. The effect of changing temperature from 19 to 24 °C on growth rate is not very detrimental (by 10 percent) compared to the effect of other manipulations of the culture medium in literature. E. huxleyi has a broad tolerance and

adaptability to temperature change compared other taxa, such as C. pelagicus.

Authors' response: Please see comment point (3).

(7) Page 6 Line 26-27: I disagree with this statement. Also Table 3 should be given the starting conditions.

Authors' response: This statement has been removed and only the DIC values measured in the cultures are left as information in the text. Starting conditions for the cultures was the growth medium and this was the same for all cultures grown in either 0.5 and 10 $\mu$M medium. Therefore, we have added this information to the methods section rather than to table 3 to preserve clarity.

(8) Page 7 Line 1-2: How about the number of layers of coccoliths forming the spheres? This could be useful to put in the context of the dynamics of cell division.

Authors' response: We agree that this would be useful information. However, it is difficult to unambiguously determine the number of layers of coccoliths under SEM. This would require a cross section of the cell, see e.g. Hoffmann et al. (2015), which is a method that was not available to us.

(9) Page 7 Lines 35-36: The Authors should add a discussion on the mechanisms for this observation. There are a few studies on cells being stuck in the haploid phase due to the lack of N and P provision to replicate DNA and allow further division.

Authors' response: Although the role of N and P in switching between haploid (non-calcified) and diploid (calcified) phases in coccolithophores is intriguing, to our knowledge no data has been published so far to unequivocally show the role of nutrient limitation in phase switches. As we did not observe phase switches or address the haploid phase in the manuscript we feel that including this theory is too far removed from the main focus of this paper. We do discuss the lack of P-provision to replicate DNA, blocking further cell division in the diploid phase, which is the reason for the cultures entering stationary phase under P-limitation.

(10) Page 8 Lines 8-9: Please refer to recent study on this particular point by Aloisi in Biogeosciences 15: 4665-4692, and incorporate suitable discussion on the mechanisms.

Authors' response: We have elaborated on the mechanisms for the cell size increase under P-limitation based on the reference suggested by the reviewer and the work of Müller et al. (2008).

(11) Page 9 Lines 6-7: I do not follow the argument being made here. Please clarify.

Authors' response: The two possible reasons for the discrepancy between coccolith numbers per cell and PIC content have now been separated into two paragraphs for clarity. The argument and discussion of partially dissolved coccoliths is now mentioned in the methods section instead of introducing it at this point.

(12) Page 10 Lines 1-2: I recommend that the Authors tone this down, as we know that such a conclusion at the scale of the global biogeochemical cycle requires longer-term and multi-strain investigation although I appreciate the "may" being used here.

Authors' response: This sentence has been reformulated.

Anonymous Referee #3: (1) They find that P-limitation can actually decrease the PIC quota, PIC production, and PIC/POC ratios in E. huxleyi, which is opposite that which has been most commonly reported in earlier literature, although some more recent studies are cited to report similar results. This contrast is little discussed.

Authors' response: Earlier experiments e.g. by Paasche used either batch or continuous (i.e. chemostat) cultures. As discussed in the text and illustrated by our data on P-limited batch cultures, there is an overriding effect of growth phase changes i.e. the change from exponential phase to stationary phase. In stationary phase, P-limitation blocks further cell division and the cells remain in the G1 (assimilation) phase of the cell cycle in which both photosynthesis and calcification continue, leading to, respectively, an increase in POC content/cell size and an overcalcification of the cells.

A similar process may be at play in chemostats in which growth is continuous, but at a lower rate. Therefore cells will be in the assimilation phase longer which may lead to an overproduction of coccoliths such as reported in Paasche and Brubak (1994). Please note that the POC content also increased in their chemostat. Similar results have been observed by Müller et al. (2008) and Perrin et al. (2016). An exception is the study by Oviedo et al. (2014) who also reported a decrease in the PIC/POC ratio in five out of six E. huxleyi strains. However, also in this study the absolute PIC content increased, together with POC content/cell size in stationary phase.

To our knowledge, ours is the first study to address nutrient limitation without introducing the confounding factor of growth rate changes. So most of this contrast can be explained by methodological differences. The relevant parts of the discussion have been modified to more clearly illustrate this issue.

(2) Neither the P-stress nor the heat stress used is very clearly justified. Where are such changes predicted to occur? Why is P-stress chosen instead of N-stress, when much more of the world's ocean is thought to show N-limitation of primary production?

Authors' response: Although much of the world's ocean is thought to be N- rather than P-limited, P-limitation can be relevant locally. This particular strain of E. huxleyi for example was isolated from the inner Oslo Fjord where the load of N over P can exceed the Redfield ratio and winter N concentrations are usually high (http://www.miljodirektoratet.no/old/klif/publikasjoner/2253/ta2253.pdf). Furthermore, P-stress may be more relevant than N-stress for calcification as P-resources are necessary for energy storage and are a part of cellular membranes, two aspects that are relevant to coccolith production.

(3) It is especially not clear to me what natural conditions are mimicked in the P-limited batch cultures. Do E. huxleyi blooms naturally experience these chemical conditions (e.g., such low DIC and omega-calcite values)? If these are conditions arise in batch cultures at very high cell densities only reachable in lab monocultures, perhaps they

must be more careful of extrapolating their results from stationary phase cultures to changes in carbon export.

Authors' response: The reviewer correctly points out that the low DIC values (due to high cell concentrations) are unlikely to occur in nature. However, we used P-limited batch cultures vs. semi-continuous cultures to test physiological limits i.e. what effect does P-limitation have once it is growth-limiting i.e. blocking further cell division vs. when cell division is continuous. Furthermore, we only extrapolated results for carbon export in the natural system from semi-continuous cultures (with more realistic cell concentrations and carbonate chemistry) and use P-limited batch cultures as a comparison.

(4) In terms of heat stress, it's not made very clear why the temperatures of 19°C and 24°C were chosen, although there is some justification given in the Discussion. Is 19°C a typical SST in the North Sea (assuming the clone here represents a North Sea population) or typical of the Oslo Fjord? With global warming is it expected to reach 24°C regularly? Authors' response: Please see common point (3). A temperature of 19°C can be considered a high summer temperature in the Oslo Fjord. As E. huxleyi often has maximum growth rates at temperatures above those found at the isolation site (Sett et al., 2014) we chose a temperature that was at the high end of the range that this strains is likely to encounter in nature as our "normal temperature".

It is unlikely that the Oslo Fjord will reach temperatures as high as 24°C regularly, at least on time scales that would preclude adaptation to higher temperature. The reason for choosing this temperature was that it was high enough to induce heat stress (defined as a decrease in growth rate) without causing the culture to crash. We were therefore focused on choosing a feasible culturing temperature that allowed us to observe the physiological effects of heat stress rather than recreating the natural environment. This has been stated up front in the methods section.

What about E. huxleyi populations currently found in 8-12°C waters, would the same

tendencies occur if grown at from 13°C to 19°C?

Authors' response: This is a difficult question to answer and would depend very much on whether the physiological range of that strain corresponds to its natural range. As the reviewer rightly cites in the specific comments, numerous studies have tested the effect of temperature on carbon production in E. huxleyi using similar or higher temperatures than we have (24°C). The temperature that is above the optimum will depend on the strain (and hence likely the place of isolation).

In this regard it is important to define "above-optimum temperature". This is defined as the temperature at which growth rate declines in respect to maximum growth rates obtained at the optimum temperature (please see common point (3)). This is precisely the reason why the temperature of 24°C did not induce heat stress in Feng et al. (2008) because the growth rate increased from 20 to 24°C. Incidentally, this strain was isolated from the Sargasso sea (strain CCMP 371) with a known temperature range of 17-26°C. Rosas-Navarro et al. (2016) report that growth rates decreased in three strains of E. huxleyi at 27.5°C. However, they do not present data for PIC and POC production, or occurrence of malformed coccoliths, at this above-optimum temperature, which is unfortunate as this would have been a very interesting data set for comparison.

(5) It's not necessary for all studies to try to replicate specific environmental conditions (often impossible), perhaps especially when the goal is to understand physiological limits or to take a first approximation.

Authors' response: As the reviewer rightly points out, our choice of experimental set-up was based on testing the physiological limits of P-limitation and heat stress, which defined also the temperatures to use as outlined above.

However, considering this lack of grounding of experimental design within an explicit environmental context, it appears that the Conclusions should be more cautious in extrapolating to biogeochemical effects.

Authors' response: We have tried to outline our experimental design/motivations more clearly in the answers to the above comments and have also clarified these aspects in the text.

(6) There is a focus on biogeochemical effects, but nothing on ecological effects of the documented changes in PIC and coccoliths. What function do they serve? I might suggest the review by Monteiro et al. to look at, and think of some of the consequences.

Authors' response: A consideration of the possible functions of coccoliths and hence consequences of lower calcification rate/coccolith coverage has been added to the discussion.

(7) Finally, I'm not so sure of the extent of the novelty of this study. They say "To our knowledge, this study is the first to specifically test the impact of heat stress..." but then there actually are a few quite relevant studies (it depends on how "heat stress" is defined), some of which they cite. For that reason, a more rigorous study design in an explicit environmental context would have been much stronger.

Authors' response: We have now defined what we mean by the term "heat stress" i.e. a decrease in growth rate, both in the replies to the above comments and in the text. Considering this definition, we are not aware that other studies have examined the effect of heat stress on PIC production.

Specific comments:

(1) p. 1 Line 26, should probably cite something more recent as well, such as the metaanalysis by Meyer and Riebesell 2015.

Authors' response: This citation has been added.

(2) p. 2 Lines 6-9 : "Batch culture on the other hand represents an end-of-bloom scenario in which the lack of nutrients limits further cell division... production cannot be determined in the batch approach ". That's only true if the last part of a batch culture is analyzed, as growth becomes limited due to exhaustion of nutrients, build-up

of metabolites, shading, limited gas exchange, etc. In fact, there are many published experiments where production rates were determined in dilute batch culture, in the early exponential phase of growth before DIC consumption or nutrient consumption was substantial.

Authors' response: The reviewer correctly points out that production rates can be determined in dilute batch cultures, in which growth is exponential. We could for example have examined the effect of heat stress in dilute batch cultures. However, it is not possible to test P-limitation in dilute batch cultures as these per definition will not be limited by nutrients such as P. We have added a sentence in the introduction to clarify this point.

(3) Line 29: "None of these studies, however, tested the effect of above-optimum temperature ". I don't understand this unless one defines what is "above-optimum temperature". In the study presented here the temperatures used are 19°C and 24°C. The study of Feng et al. (2008) (which they cite) used 20°C and 24°C, so if 24°C is an above-optimum temperature in the present study, why wasn't it in Feng et al.? Rosas-Navarro et al. 2016), which they also cite, used 25°C as the highest temperature. How is "above-optimum temperature" defined, and wouldn't it possibly depend on the origin of the cultures? For example, E. huxleyi from lat. 50°C in the North Atlantic probably does not experience such temperature, E. huxleyi in the Mediterranean Sea will rarely experience temperatures that high, and those may be unexceptional surface temperatures for the tropical Pacific, where E. huxleyi can also be found. This is discussed much later on p. 8, lines 20-26, but I am not so convinced how relevant this temperature range is.

Authors' response: Please see common point (3).

(4) Lines 36-37: Should consider (and cite) also work of van Bleiswijk et al. 1994 and Rokitta et al. 2016 very relevant for theme of E. huxleyi response to P-limitation.

Authors' response: These references have been considered in the final version.

(5) I have a problem with the use of K medium for nutrient experiments. K medium contains a mix of ammonium and nitrate as N-source, and it contains glycerophosphate as a P source. It's not clear from Gerecht et al. (2014) if they modified these components. They need to give the basal medium composition they used.

Authors' response: It is correct that K-medium usually contains glycerophosphate as a P-source, as well as ammonia. We, however, modified the medium to contain only nitrate as a N-source and KH2PO4 as a P-source. The full medium recipe has been added to the methods section.

What volume were cultures?

Authors' response: Batch cultures were 350 mL, whereas semi-continuous cultures were kept at 50 mL until the last dilution round where volumes were increased to 350 mL.

(6) p.4 For semi-continuous cultures, what was the dilution rate or growth rate?

Authors' response: The average growth rates for semi-continuous cultures are reported in Table 2. The dilution rate varied slightly depending on the cell concentrations reached in the cultures after two days as all cultures were diluted back down to 10.000 cells mL-1 every two days. A supplementary figure has been added to show the development of cell concentrations for each dilution cycle of semi-continuous cultures.

How was it determined or confirmed that the cultures in fact were limited by P-limitation in the semi-continuous cultures?

Authors' response: During the course of the experiment we could not confirm whether the cultures were limited, as the growth rate was not affected. We could confirm that the cultures were limited only after harvesting the cultures and determining phosphorus (POP) content which was lower in cultures grown on 0.5 $\mu$M initial phosphate medium than at 10 $\mu$M.

How could maximum cell concentrations have reached 170000 cells/ml if cultures were

diluted back to 10000 cells/ml every second day? To have reached 170000 cells/ml from 10000 cells/ml in only 2 days would require aprox. 4 cell divisions per day, which has never been reported for this species.

Authors' response: To reach a maximum of 170.000 from 10.000 cells mL-1 in two days requires approx. 4 cell divisions in two days i.e. approx. 2 cell divisions per day, which is in the range of what has been reported for this species.

(7) Line 7: "P-limited cultures were harvested in stationary phase, . . ." for how long in stationary phase? This is not clear from Fig. 4.

Authors' response: P-limited cultures were harvested on the day of the last data point presented in Fig. 4. We have added arrows to the figure to illustrate this.

(8) p. 5 Line 10: Give manufacturer & city for "CASY".

Authors' response: Manufacturer & city are given at first mention of the instrument (p. 4 line 6).

(9) Line 30: "Average values were compared by a t-test". Was this pairwise test performed after the two-way ANOVA? If so, with what correction for multiple comparison? They are testing two factors (T and P-limitation) so should be doing a two-way ANOVA, not t-tests.

Authors' response: The reviewer rightly points out that a two-way ANOVA is necessary to test the effect of two factors (T and P-limitation). We used t-tests when comparing the effect of only one of these factors. However, as we describe both factors together in the text, the reference to the t-test is obsolete and has been removed. As stated in the next sentence, we used a two-way ANOVA to compare the data ("The influence of P-availability and temperature on variables was determined by means of a two-way analysis of variance (ANOVA).").

(10) p. 6 Line 24 and Table 3: What limited the growth of control batch cultures?

Authors' response: Nothing was limiting control batch cultures as they were harvested in exponential phase.

(11) p. 8 Lines 10-12: "These large "ready-to-divide" cells (Gibbs et al., 2013) not only accumulate POC, but also accumulate PIC, leading to the 2-3-fold increase in coccolith number per cell observed in stationary phase cultures (Fig. 2c,d)." How do you know these cells are "ready-to-divide"? If they really are "ready-to-divide", do you mean they are blocked in G2 or M phase of the cell cycle? That doesn't make much sense.

Authors' response: This sentence has been rephrased.

(12) Lines 10-12: This is an important justification for their selection of temperatures. Nevertheless, I'm not very convinced about how these temperatures aare reoe. I would prefer them to explicitly give the range of temperatures experienced in the North Sea as well as the fjord. Why is 19 °C a "normal temperature"? What does that mean?

Authors' response: The natural temperature range that can be expected for this strain has been added to the text. As to the definition of the temperatures used please see common point (3).

(13) Lines 12-13: "Stationary phase can be likened to an end-of-bloom scenario in nature, during which E. huxleyi sheds numerous coccoliths, leading to the characteristic milky color of coccolithophore blooms". Maybe, but it's also well know that the end of E. huxleyi blooms involves infection by the virus EHV.

Authors' response: In this statement we are not referring to what causes the demise of E. huxleyi blooms in nature (for which there may be numerous reasons), but rather describing the characteristic "overproduction" of coccoliths.

(14) p. 9 Line 10 "The percentage of partially dissolved coccoliths was higher at normal temperature than under heat" Where is this shown? Data is presented on "incomplete", "malformed", and "normal" coccoliths in Fig. 3. They state "The high numbers of incomplete coccoliths observed in P-limited batch cultures were likely a result of secondary

dissolution (Fig. 1d; Langer et al., 2007) due to the low calcite saturation state reached in stationary phase cultures." I would like to see more examples of incomplete coccoliths. Perhaps they can show that the type of incompleteness that appears in P-limited batch cultures (when omega-calcite is less than 1) is distinct from what appears when omega-calcite is greater than one?

Authors' response: Please see common point (2).

Anonymous referee #4:

(1) I feel that the discussion is weakened by an emphasis on comparing two culturing methods rather than comparing individual vs. interactive effects of the stressors in question.

Authors' response: We would like to point out that the inclusion and discussion of the two culturing methods in the manuscript not only serves as a "methodological comparison". It also serves to compare two differing environmental scenarios. Whereas batch culture represents a strong P-limitation as may be encountered at the end of blooms, the semi-continuous culture tests the effect of a more continuous low-P environment. We agree that the discussion would benefit from a comparison of individual vs. interactive effects of the two stressors and this has been added.

(2) The specific choice of experimental conditions is also poorly justified and not placed into context.

Authors' response: Please see common point (3) and reply to general comment (2) of referee #3.

Specific Comments:

(1) In the broader interpretations of their calcification results, the authors state in the Abstract, Introduction and Conclusions that decreases in calcification rates in E. huxleyi could "lessen the ballasting effect of coccoliths and weaken carbon export out of the photic zone", or similar wording. In support, they reference Ziveri et al. (2007), who

conclude that, despite the high abundance of Emiliania relative to other coccolithophore species, the small size and very low species-specific carbonate mass of their coccoliths means that they consequently export far less carbonate than expected. Baumann et al. (2004) similarly concluded that Emiliania plays only a relatively minor role in carbonate export in the Equatorial and South Atlantic. Can the authors support their statements of alterations to the carbon cycle more quantitatively using their PIC production values and abundances of Emiliania in the field? Or provide references of studies that better support these statements?

Authors' response: The reviewer correctly points out that the overall contribution of E. huxleyi to pelagic carbonate flux is small compared to other species. Nevertheless, coccolith ballasting (or lack thereof) can be considered relevant locally e.g. during blooms of E. huxleyi. To support the above statement i.e. weaker ballasting due to decreased calcification rates, it is necessary to examine whether and to what extent these physiological responses are applicable to coccolithophores as a whole. We have some indication that other species react similarly, including those arguably more (regionally) relevant to carbonate export such as C. pelagicus. For example, we have previously reported that C. pelagicus increases P-requirements and decreases coccolith coverage under heat stress (Gerecht et al., 2014). The relevant parts of the text have been modified to reflect this.

(2) Considerable emphasis is made on the short-comings of the batch culturing technique compared to the semi-continuous culture technique. Comparison of these two methods is present throughout the results and discussion, and, in my opinion, obscures a clear and explicit evaluation of contrasting individual (warming only, P-limitation only) vs. interactive (warming and P-limitation) effects and evidence (or not) for positive interactions.

Authors' response: Please see reply to general comment (1).

It is stated that batch culture experiments can only represent a severely nutrient depleted scenario whereas a semi-continuous set-up provides an acclimated low-nutrient population (p3, ln 7; p7 lns 12-17). I think that these statements are somewhat misleading for the following reason: A batch culture experiment experiences exponential growth at whatever the starting nutrient concentrations until these nutrients become sufficiently depleted that exponential rates of growth can no longer be maintained and growth rate rapidly falls to zero. A semi-continuous culture is just a batch culture that is subcultured/diluted (typically) around mid-exponential-phase cell concentrations several times. I therefore find it strange that the authors did not just sample their P-limited batch culture experiment at mid-exponential phase (as they did with the control experiments) well before the 'severe' nutrient depletion of stationary phase began, which based on Fig. 4 would have meant sampling on day 3 or 4. Would this not have been a more realistic comparison of control and P-limited conditions during exponential phase in all experiments?

Authors' response: Although the reviewer is correct insofar as there are similarities between an exponential batch culture and a semi-continuous culture, these two scenarios are not identical. The crucial difference is that a semi-continuous culture experiences stable limiting conditions over many generations, which leaves ample time for acclimation processes to re-structure the physiological machinery dealing with this environmental stress. In a batch culture, by contrast, the specific state of limitation equivalent to a semi-continuous scenario is a transient state experienced by the cells for a short time only (less than one generation). The whole point of comparing batch and semi-continuous culture was to compare the cumulative effect of a series of transient limitation states (increasingly severe; batch) to a single constant limitation state representing roughly the average of the many transient batch culture states (semi-continuous). Our experimental setup was designed to serve this purpose.

It would also allow a comparison of multiple generations of growth (semi-continuous) with far fewer generations of growth (batch to mid-exponential phase).

Authors' response: We do not see the benefit of this.

It would also have removed the issues with calculating production rates, as only exponential growth phase populations would have been sampled.

Authors' response: True, but see comments above.

(3) I would like to see a broader context of the area where a similar degree of temperature and phosphorus stress is predicted to be experienced in the context of this strain isolated from a Norwegian fjord.

Authors' response: The aim of this study was not to test/evaluate specific predicted environmental factors relevant for our strain. The aim was to look at environmental stressors (high temperature, P-limitation) by testing the physiological limits, independently of whether and when this strain will encounter these conditions in nature. Therefore we chose a temperature that was above the optimum for growth (but that still allowed growth; please see common point (3)) to test the effect of heat stress. Similarly we tested the effect of strong and weak P-limitation on this strain by having one laboratory scenario in which P-limitation becomes limiting for cell replication (stationary phase batch culture) and one in which P-limitation is not strong enough to affect cell replication i.e. growth rate, but does affect calcification rate (semi-continuous culture). In this regard, see also reply to general comment (3) of referee #1 and to general comments (2) and (4) of referee #3.

Similarly, there is no discussion of the fact that physiological stress experience by one strain of this species under climate change is as likely to lead to its ecological replacement by another, more tolerant strain given recent studies presenting the large genetic pool of Emiliania (e.g., Read et al., 2013, which also discusses differences in genes for tolerance of low phosphorus conditions between strains).

Authors' response: This aspect has now been considered in the discussion.

(4) I was surprised to see that no figures of any POC, PIC, or POP data were presented, only data in the tables. Was there a reason for this? Given the two experimental

approaches, two temperatures and two nutrient states, it made it difficult to quickly visualise the dataset.

Authors' response: There was no particular reason, apart from keeping the manuscript concise. We will add the relevant figures as a supplement to the final version.

(5) The strain used seems to be a new isolate – do the authors intend to deposit this strain into a culture collection for use by other researchers? Given it is not held in a culture collection, the authors must provide the essential ancillary information on the isolate and its maintenance in culture.

Authors' response: Please see common point (1).

(6) It is stated deep into the discussion that the temperature at the isolation location does not exceed 21 degrees and this is presumably how the authors decided that a temperature of 24 degrees was beyond the thermal optimum. Did the authors perform a systematic temperature optimum assessment by determining growth rates at a range of temperatures?

Authors' response: Please see common point (3).

Given that the exponential growth rates were not substantially different between temperature treatments (semi-continuous) and in fact were higher in the 24 degree treatment for the batch culture approach, this would perhaps suggest that (using exponential growth rate as a physiological indicator) this isolate has a relatively broad thermal tolerance.

Authors' response: Please see common point (3).

(7) The authors do not state whether there was any period of acclimation for populations experiencing low phosphate or high temperature treatments.

Authors' response: Cultures were acclimated for ca. 10 generations to low phosphate and high temperature culture conditions before starting the experiment. This information has been added to the methods section.

(8) How did the authors account for the tendency of Emiliania to form multi-layer coccospheres when counting the number of coccoliths from SEM images? Comparing the coccosphere size from CASY with the cell size from light microscopy (back calculated from the volume data) and considering the thickness of Emiliania coccoliths, would suggest that coccospheres were not mono-layer.

Authors' response: The majority of multilayered coccospheres found in our study collapsed during the filtration process, which allowed us to count the coccoliths from all layers. The non-collapsed multilayered coccospheres commonly had an only partially covered first layer i.e. an incomplete second layer. Thus, it was possible to count the visible coccoliths in the first and second layer and estimate the number of covered coccoliths of the first layer. In very rare cases when we could not estimate the number of layers due to a complete outer layer, we estimated the number of layers and the number of coccoliths, which could fit under the outermost layer. This approach was consistently used throughout the SEM analysis to minimize error arising from the inability to see all coccoliths on the coccosphere.

(9) Is there any reason why the authors refer to both light microscope and CASY cell size measurements as "cell size" when CASY gives coccosphere size measurements? There are huge differences in volumes between the two methods due to the coccosphere and whilst cell size directly relates to cell carbon, coccosphere size does not and therefore this unnecessarily confuses the reading in parts.

Authors' response: The CASY system does not actually provide accurate coccosphere size measurements (Gerecht et al., 2015), but gives an intermediate value between cell and coccosphere size. We therefore use CASY measurements as a proxy for "cell size" to be able to observe and visualize the size increase during the development of the batch culture.

(10) The discussion would benefit from a discussion of the physiological mechanisms

behind the observed response to P-limitation and heat stress singularly and combined, and there is considerable literature on this species, other coccolithophore species and other phytoplankton groups that would support such a discussion.

Authors' response: We do discuss the possible physiological mechanism behind the cell size increase under P-limitation, as well as the possible explanations for the increased P-requirements under heat stress. As this is the first observation of decreased calcification rate under heat stress and P-limitation in this species, we find it difficult to speculate further about the possible physiological mechanisms unless the referee would like to point us towards relevant literature. In this regard, please also see reply to general comment (4) of referee #2.

p6, ln 37-39 – referred to changes in size but "twice as large in stationary phase" refers to cell volume, so this should be changed to reflect this.

Authors' response: The reference to cell volume has been added.

p8, ln 7-9 – The reference to Sheward et al. (2016) on line 7 should be changed to Gibbs et al. (2013) as the latter presented Emiliania data. This sentence could also include C. braarudii, Calcidiscus and Helicosphaera from Sheward et al. (2017). The Sheward et al. (2016) at the end of line 9 should be changed to 2017 (I think you have referred to the discussion paper rather than the finally-published article).

Authors' response: The relevant references have been changed.

Technical Corrections:

Throughout the paper, there are inconsistencies with the author order in your references. Sometimes they are ordered by date, other times alphabetically, and several times I can find no logic to the order! (e.g., p2, ln 37-38).

Authors' response: This has been resolved.

Anonymous referee #5:

. . .the choice of the cultures conditions, P limitation and temperature stress, need to be express with more details in the introduction or in the methodology part concerning the choice of the values.

Authors' response: Please see common point (3).

Specific comments

p. 2; Ln 9: You need to add the word particulate to present for the first time in your manuscript the PIC and POC terms.

Authors' response: This has been added.

p. 3; 19: What does inter alia mean?

Authors' response: Latin for "among other things"; "inter alia" has been replaced with this term in the text.

p. 3; Ln 29 & 35 : Did you test different temperature on the strain (before these experiment) to find the optimal temperature at the selected light dose (12 :12, 100_mol photons.m-2.s-1)? Or did you have an idea (or reference) of the optimal temperature of your strain before selecting your two temperature conditions?

Authors' response: Please see common point (3).

p.3; Ln 34-39: What is the name of your strain?

Authors' response: Please see common point (1).

Have you done some acclimations of your strain to the temperature and P concentration conditions before starting your experiment? You should indicate in your method if you did.

Authors' response: Yes, cultures were acclimated for ca. 10 generations to the temperature and P concentration conditions before starting the experiment. This information has been added to the methods section.

What is the initial nitrate concentration in your medium? Is it the standard concentration of NO3 in a K medium? If not, the value of the concentration need to be indicate in this method part.

Authors' response: The basal composition of the medium has been added to the methods section.

You also need to add the value of the salinity of your medium.

Authors' response: This information has been added to the methods section.

p. 5; Ln 21-26: Did you take into account that Emiliania huxleyi can have multiple layers of coccoliths? It is clearly visible on your figure 1 that your strain can have several layers of coccoliths depending of the cultures conditions. How did you take this particularity in your coccoliths counts?

Authors' response: Please see reply to specific comment (8) of referee #4.

What is the standard deviation on your counts? At least, you have triplicate so you need to specify the standard deviation on your number.

Authors' response: The standard deviation of our counts of coccolith number per cell is listed for each culture condition in Table 1. This standard deviation is based on the total number of cells (n) analysed, which is the sum of all three replicate cultures for each condition. N is also listed for each culture condition in Table 1.

What about coccospheres diameter on your SEM images? Did you think about doing coccopheres measurements? If not, is it because of the high number of detached coccoliths on your filter?

Authors' response: As the reviewer points out, there was a high number of detached coccoliths on the filter so that coccosphere measurements under SEM would possibly have been underestimated. Additionally, most of the coccospheres in P-limited batch cultures had collapsed, making accurate estimates of coccosphere size based on SEM

measurements difficult.

It should be interesting to discuss the PIC content related to your coccolith number and the thickness of your coccolith layer.

Authors' response: Yes, we agree with the reviewer that this would have been an interesting aspect to examine. However, determining the thickness of the coccolith layer or the number of coccolith layers around one cell with SEM was not possible; see also response to specific comment (8) of referee #2.

p. 6; Ln 24-34: Final cell density for batch cultures were being push really high in order to get the P-limitation. However, respect to LaRoche et al. 2010, inorganic carbon system changes need to be kept below 5% to avoid carbon system changing due to the high cell density. This meens that nutrient limitation experiment for batch system have to attain a reasonable final cell density in order to keep a DIC system quasiconstant through the experiment and to avoid a inorganic carbon limitation before the P limitation in this case. In your P-limited cultures, DIC changes that you get at the end of the experiment led to a low calcite saturation constant. Therefore observations in morphology and calcite content may be due to this changes rather than the P-limitation and heat stress effect. You took into account this changes in your discussion but it will be useful to indicate why you did batch experiments in this way. Could you justify this choice in the methodology of your batch experiment? Did you think about carrying the experiment with a lower initial P concentration rather than push the cell density so high? Or did you have a target initial P concentration that you wanted to test? In this last case, a comment on the 2.1 part will be necessary.

Authors' response: We did not choose an initial phosphate concentration of 0.5 $\mu$M to recreate a specific field situation. Rather, this concentration was chosen based on prior considerations of collecting enough biomass for all analyses, while keeping cell concentrations of semi-continuous cultures well below stationary phase. The low DIC concentrations in P-limited stationary phase batch cultures do no affect the conclusions

of the manuscript in regard to P-limitation, which are that P-limitation does not affect coccolith morphology per se, but decreases calcification rate in this strain of E. huxleyi.

p. 7; Ln 27: What is the normal temperature? Even if you explain it later (p.8 Ln 26), you should describe here that the normal temperature is 19°C if you want to use this term. It is not obvious for readers.

Authors' response: The terms (and motivations) of using 19°C as normal and 24°C as supraoptimal temperature have been now explained in the methods section.

p. 8; Ln 1-5: same comments that previously on the DIC system

Authors' response: See response to previous comments.

Figures & Tables

Figure 2: You need to add the color explanations: blue is your schematic initial cell, red is the schematic effect observed at the end of your experiment.

Authors' response: This information has been added.

Figure 4: You should add the error bar on triplicate somewhere in your legend.

Authors' response: This information has been added.

Tables 1, 2 & 3: You should add the standard deviation in your legend and your n number (for example, n = 3 if it is triplicate).

Authors' response: This has been added.

Table 2: You need to clarify in the legend of your table if the cell volume has been calculated with measurements of cell diameter from the harvest day or from an average of daily measurements.

Authors' response: The cell volume presented in the table was calculated from measurements of cell diameter from the harvest day. This information has been added to the table.

Cited references

Aloisi, G.: Covariation of metabolic rates and cell size in coccolithophores, Biogeosciences, 12, 4665-4692, 2015.

Borchard, C., Borges, A. V., Händel, N., and Engel, A.: Biogeochemical response of Emiliania huxleyi (PML B92/11) to elevated CO2 and temperature under phosphorus limitation: a chemostat study, J. Exp. Mar. Biol. Ecol., 410, 61-71, 2011.

Buitenhuis, E. T., de Baar, H. J. W., and Veldhuis, M. J. W.: Photosynthesis and calcification by Emiliania huxleyi (Prymensiophyceae) as a function of inorganic carbon species, J. Phycol., 35, 949-959, 1999.

Eppley, R. W.: Temperature and phytoplankton growth in the sea, Fishery Bulletin, 70, 1063-1085, 1972.

Feng, Y., Warner, M. E., Zhang, Y., Sun, J., Fu, F.-X., Rose, J. M., and Hutchins, D. A.: Interactive effects of increased pCO2, temperature and irradiance on the marine coccolithophore Emiliania huxleyi (Prymnesiophyceae), Eur. J. Phycol., 43, 87-98, 2008.

Gerecht, A. C., Šupraha, L., Edvardsen, B., Probert, I., and Henderiks, J.: High temperature decreases the PIC / POC ratio and increases phosphorus requirements in Coccolithus pelagicus (Haptophyta), Biogeosciences, 11, 3531-3545, 2014.

Gerecht, A. C., Šupraha, L., Edvardsen, B., Langer, G., and Henderiks, J.: Phosphorus availability modifies carbon production in Coccolithus pelagicus (Haptophyta), J. Exp. Mar. Biol. Ecol., 472, 24-31, 2015.

Hoffmann, R., Kirchlechner, C., Langer, G., Wochnik, A. S., Griesshaber, E., Schmahl, W. W., and Scheu, C.: Insight into Emiliania huxleyi coccospheres by focused ion beam sectioning, Biogeosciences, 12, 825-834, 2015.

Langer, G., Nehrke, G., Probert, I., Ly, J., and Ziveri, P.: Strain-specific responses of

[Figure]

Emiliania huxleyi to changing seawater carbonate chemistry, Biogeosciences, 6, 2637-2646, 2009.

Langer, G., Oetjen, K., and Brenneis, T.: Coccolithophores do not increase particulate carbon production under nutrient limitation: a case study using Emiliania huxleyi (PML B92/11), J. Exp. Mar. Biol. Ecol., 443, 155-161, 2013.

Müller, M. N., Antia, A. N., and LaRoche, J.: Influence of cell cycle phase on calcification in the coccolithophore Emiliania huxleyi, Limnol. Oceanogr., 53, 506-512, 2008.

Oviedo, A. M., Langer, G., and Ziveri, P.: Effect of phosphorus limitation on coccolith morphology and element ratios in Mediterranean strains of the coccolithophore Emiliania huxleyi, J. Exp. Mar. Biol. Ecol., 459, 105-113, 2014.

Paasche, E., and Brubak, S.: Enhanced calcification in the coccolithophorid Emiliania huxleyi (Haptophyceae) under phosphorus limitation, Phycologia, 33, 324-330, 1994.

Paasche, E., Brubak, S., Skattebøl, S., Young, J. R., and Green, J. C.: Growth and calcification in the coccolithophorid Emiliania huxleyi (Haptophyceae) at low salinities, Phycologia, 35, 394-403, 1996.

Paasche, E.: Roles of nitrogen and phosphorus in coccolith formation in Emiliania huxleyi (Prymnesiophyceae), Eur. J. Phycol., 33, 33-43, 1998.

Perrin, L., Probert, I., Langer, G., and Aloisi, G.: Growth of the coccolithophore Emiliania huxleyi in light- and nutrient-limited batch reactors: relevance for the BIOSOPE deep ecological niche of coccolithophores, Biogeosciences, 13, 5983-6001, 2016.

Riegman, R., Stolte, W., Noordeloos, A. A. M., and Slezak, D.: Nutrient uptake and alkaline phosphatase (EC 3:1:3:1) activity of Emiliania huxleyi (Prymnesiophyceae) during growth under N and P limitation in continuous cultures, J. Phycol., 36, 87-96, 2000.

Rosas-Navarro, A., Langer, G., and Ziveri, P.: Temperature affects the morphology and

calcification of Emiliania huxleyi strains, Biogeosciences, 13, 2913-2926, 2016.

Satoh, M., Iwamoto, K., Suzuki, I., and Shiraiwa, Y.: Cold stress stimulates intracellular calcification by the coccolithophore, Emiliania huxleyi (Haptophyceae) under phosphate-deficient conditions, Mar. Biotechnol., 11, 327-333, 2009.

Sett, S., Bach, L. T., Schulz, K. G., Koch-Klavsen, S., Lebrato, M., and Riebesell, U.: Temperature modulated coccolithophoris sensitivity of growth, photosynthesis and calcification to increasing seawater pCO2, PLOS ONE, 9, e88308, 2014.

van Bleijswijk, J. D. L., van der Wal, P., Kempers, R., Veldhuis, M. J. W., Young, J. R., Muyzer, G., de Vrind-de Jong, E. W., and Westbroek, P.: Distribution of two types of Emiliania huxleyi (Prymnesiophyceae) in the Northeast Atlantic region as determined by immunofluorescence and coccolith morphology, J. Phycol., 27, 566-570, 1991.

Wördenweber, R., Rokitta, S. D., Heidenreich, E., Corona, K., Kirschhöfer, F., Fahl, K., Klocke, J. L., Kottke, T., Brenner-Weiß, G., Rost, B., Mussgnug, J. H., and Kruse, O.: Phosphorus and nitrogen starvation reveal life-cycle specific responses in the metabolome of Emiliania huxleyi (Haptophyta), Limnol. Oceanogr., 10.1002/lno.10624, 2017.

---

## Author Response (AR1)

Point-by-point reply to reviewers' comments:

**Anonymous Referee #1:**
*1) The environmental significance of the data is presented with appropriate caution in the main text, but becomes over-stated in the abstract and conclusions sections.*
We agree that general statements on the effects of P-limitation and heat stress on *E. huxleyi* or coccolithophores as a whole, require further investigation of additional strains and/or species to account for strain- and species-specific variability. We have modified the relevant section of the abstract (coccolithophores now reads *E. huxleyi*) and conclusions (*E. huxleyi* now reads "a temperate strain of *E. huxleyi*") to specify that our interpretations are specific to *E. huxleyi*/the strain under study. We have also elaborated on which physiological effects may be more general e.g. an increase in phosphorus requirements at higher temperature, observed also for *Coccolithus pelagicus* (Gerecht et al., 2014) and inferable from the data presented by Satoh et al. (2009) and Feng et al. (2008) for two additional strains of *E. huxleyi*. Whereas a decrease in coccolith coverage at high temperature has also been observed for *C. pelagicus* (Gerecht et al., 2014), to our knowledge this is the first study to describe a decrease in calcification under weak (i.e. not affecting growth rate) P-limitation. So further studies on other strains and species are necessary to confirm how widespread this physiological response is. We have stated this accordingly in the conclusions.

*Further recognition of the potential for acclimation is also important.*
Cultures were acclimated to experimental conditions (temperature, P-concentrations) for ca. 10 generations before starting the experiment (this has been added to the methods section). Ten generations should be sufficient for acclimation to experimental conditions, according to the model of Aloisi (2015). Longer-term adaptation to changing environment is a factor that needs to be considered and we have included a short discussion about the potential for adaptation in the discussion.

*2) The criteria for classifying coccolith morphology as 'normal, incomplete, and malformed' should be included in Section 2.4 (Methods). At present, the significance of 'incomplete' coccoliths as those that have undergone secondary dissolution (rather than being 'incomplete' due to incomplete primary formation) is only discussed in Section 4.2. This is an important distinction, particularly for fellow scientists who attempt to apply the same morphologic criteria in other experiments. An additional image of a representative coccosphere from the cultures that had higher levels of malformation would also be a useful addition to Fig. 1.*

We have added a table to the methods section to outline the characteristics of the three morphological categories "normal, incomplete, and malformed". We have also added an upfront description in the methods section of the differences between the two possible origins of "incompleteness" of coccoliths and state that we could not distinguish between them. We have added additional images to Fig. 1 to illustrate the

different classes of coccoliths in more detail and have included a representative coccosphere with high levels of malformed coccoliths (new Fig. 2e).

Other minor comments:
5    *- Table 2 (page 5 line 16) is referred to before Table 1 (page 5, line 38)*
The numbering of the tables has been changed.

*- The significance of the red and blue colors in Figure 2 is missing*
This information has been added to the figure legend.

*- There is no reference to the error bars in Figure 4.*
Reference to the error bars has been added to the figure legend.

**M. Hermoso, Referee #2:**
15    *(1) More information is needed on the cultured strain of Emiliania huxleyi, including (where possible) the date of isolation, the morphotype of coccoliths, whether the strain is deposited in a Culture Collection (or in the process to be), and the conditions under which the stock culture is maintained in the laboratory (temperature, light irradiance, etc), is the strain axenic?*
20    Additional information on the origin and isolation of this strain has been added to the methods section, including date of isolation and the conditions under which the stock culture is maintained in the laboratory. The strain is non-axenic and belongs to morphotype A, which is widespread in the Northeast Atlantic. It has been deposited at the NIVA culture collection (niva-cca.com). Unfortunately, the strain has apparently
25    ceased to calcify after deposition at the culture collection.

*(2) There are no details given of the culture technique per se apart from strategy (batch vs. semi-continuous) adopted. Were the cells acclimated to the target phosphorus concs and temperature conditions when proper experiments began?*
30    Cultures were acclimated for ca. 10 generations to the two phosphate concentrations and temperatures before starting the experiments. This information has been added to the methods section.

*(3) It would be valuable to elaborate on the malformations of the*
35    *coccospheres/coccoliths observed by the Authors. The rationale behind the discrete class of malformation features and the implications for biomineralisation are elusive in the manuscript. ... Likewise, it is not entirely clear to me how the Authors are able to distinguish between malformation and dissolution features.*
We have defined three categories of coccolith morphology: normal, malformed and
40    incomplete. We have added a table to the methods section to better outline the characteristics of these categories. We do not differentiate between different kinds of malformations, only between incomplete (considered to be of normal morphology, but are uniformly unfinished) and malformed (of overall irregular morphology or with irregular morphological features). We could clearly distinguish between incomplete

and malformed coccoliths on the SEM images. Calcite content of single malformed coccoliths can be expected to be lower, the same or higher than normal coccoliths depending on the kind of malformations, whereas incomplete coccoliths can be expected to contain less calcite (though we did not measure this).

*On this note the argument that 24°C represents a heat stress for this strain as it was isolated from waters measured at 21°C or lower, and that more malformations were observed (p. 8 lines 23-25) does not appear as a strong argument to me.*
The main evidence that these cultures were heat-stressed is provided by the decrease
10    in growth rate at 24°C compared to 19°C. The isolation temperature and increased presence of malformations at 24°C are additional indications that this temperature was above optimum. This has now been mentioned in the methods section and specified in the discussion.

15    *(4) I feel that at places the discussion is too descriptive and lacks a better attempt to understand the cellular mechanisms at play for the environmentally-driven change in carbon fixation. An integration of P acquisition strategy by E. huxleyi (a species with the ability to excrete ligands to increase P supply to the cell) with growth dynamics and organic and inorganic carbon fixation for each condition would be extremely*
20    *useful and add value to the paper.*
The reviewer points out that *E. huxleyi* has a particularly high capacity for obtaining P from its environment, e.g. Riegman et al. (2000). This characteristic makes it even more intriguing that a weak P-limitation, as imposed by our semi-continuous set-up, in which P is still readily available to the cells, should have an effect on the
25    calcification rate. The few studies that have addressed the effect of P-limitation in a continuous setup observed either an increase in calcification (Riegman et al., 2000; Paasche, 1998; Paasche et al., 1996) or no change (Borchard et al., 2011). Please note that an increase in calcification was observed in all studies in co-occurrence with a decrease in growth rate in the continuous cultures so there is the overriding effect of
30    growth rate. To our knowledge we present, for the first time, growth rate independent data on changes in calcification in *E. huxleyi* to show that calcification can decrease under P-limitation. We have some mechanistic understanding of the increase in calcification in stationary phase, mostly based on the work of Müller et al. (2008) i.e. cells are kept in the G1 (assimilation) phase of the cell cycle as P is lacking for cell
35    division and therefore coccolith production (and to some extent also POC production) continues. However, we find it difficult to speculate about possible cellular mechanisms regarding the decrease in calcification under weak P-limitation as there is no knowledge base in the literature to explain why calcification would have specific P-requirements. On the contrary, most data point to a stronger dependence of POC
40    than PIC production on P-resources, as exemplified by the increase in PIC/POC in stationary phase cultures. We elaborate on this in the revised discussion.

Specific comments:
*(1) Page 3 Lines 6-7: In my opinion, looking at Table 3 it is not nutrient limitation*

*that limits further algal growth in this set-up, but rather the drift in the carbonate chemistry of the medium (see e.g. Hermoso 2014 in Cryptogamie, Algologie 35(4): 323-351).*

If the reviewer is referring to the change in pH as "drift in the carbonate chemistry" than the change in pH was not great (lowest pH=7.7) due to the opposing effects of photosynthesis and calcification on culture medium pH. This pH-value is unlikely to have affected growth rates. If the reviewer is referring to the decrease of DIC as "drift" than DIC is also a nutrient that may have been limiting at the end of the experiment. However, although P-limited batch cultures at the time of harvest were possibly co-limited by P and DIC as outlined in the text, entry into stationary phase was due to P-limitation for the following reasons. There was a drastic decrease (3.3-3.7-fold; Table 2) in the POP content of "P-limited stationary phase cultures", which is a sign of strong P-limitation and would not be expected under DIC limitation. Secondly, POC fixation continued in stationary phase leading to a strong increase in cell size/POC content, which again argues against DIC limitation. The recent publication by Wördenweber et al. (2017) gives mechanistic support for this observation. By analyzing the metabolome, these authors observed that under P-starvation metabolites such as lipids are accumulated i.e. enzymatic functionality is preserved. This continuing fixation of POC by non-dividing cells led to the high consumption of DIC in P-limited batch cultures. We have elaborated on this for the final discussion.

*(2) More broadly, I do not believe that the stationary phase represents an end-of-bloom scenario.*

It is difficult to recreate natural situations in the laboratory and we do not pretend to have faithfully recreated an "end-of-bloom" scenario, as bloom demise is likely regulated by many factors. However, "stationary phase batch cultures" are the closest approximation we could achieve in the laboratory to an "end-of-bloom" scenario and it would now be interesting to test this hypothesis in the field.

*(3) Page 3 Line 19: inter alia?*

This expression has been changed to "among other things".

*(4) Page 3 Line 22: Carbon "fixation" rather than "production".*

Carbon production is a technical term referring to the calculation of production based on cellular elemental content (e.g. POC) and growth rate, e.g. Langer et al. (2013). This term is widely used in the literature and is the term we are referring to here.

*(5) Page 3 Lines 22-24: There are many other references (Bollmann et al. 2010 in Protist 161:78–90; McClelland et al. 2016 in SciReport 6:34263 etc etc of which some cited at the end of the discussion should be also mentioned here).*

Although numerous studies, such as those mentioned by the reviewer, have studied the effect of temperature on carbon production and coccolith shape and size, only a handful of studies have specifically examined the effect of temperature on coccolith

malformations (as defined in this manuscript). We have specified that we are referring to this aspect of temperature effects in the text.

*(6) Page 3 Line 30: I still think that "heat stress" is not appropriate here for the reasons outlined in general comments. The effect of changing temperature from 19 to 24 °C on growth rate is not very detrimental (by 10 percent) compared to the effect of other manipulations of the culture medium in literature. E. huxleyi has a broad tolerance and adaptability to temperature change compared other taxa, such as C. pelagicus.*

Yes, the decrease in growth rate at 24 compared to 19°C is slight (6-9%, new Table 4), and only observed in semi-continuous, not batch cultures. However, measurements of growth rate in semi-continuous cultures are more robust because growth rate is measured as an average of numerous dilution cycles. A decrease, even if slight, in growth rate at a higher temperature is the definition of above-optimum growth. According to Eppley (1972), phytoplankton growth rates increase with increasing temperature as long as this temperature is below the optimum for growth, whereas above optimum temperature, growth rates decline. As growth rates decline more sharply at above-optimum temperatures (i.e. heat stress is more detrimental than below-optimum temperature), it is difficult to culture phytoplankton at above-optimum temperature. Therefore, obtaining a stronger difference in growth rate at even higher temperature would have been technically difficult and would likely have resulted in the crash of the culture. These motivations and definitions have now been mentioned in the methods section.

*(7) Page 6 Line 26-27: I disagree with this statement. Also Table 3 should be given the starting conditions.*

This statement has been removed and only the DIC values measured in the cultures are left as information in the text. Starting conditions for the cultures was the growth medium and this was the same for all cultures grown in either 0.5 and 10 μM medium. Therefore, we have added this information to the methods section (new table 1) rather than to table 3 to preserve clarity.

*(8) Page 7 Line 1-2: How about the number of layers of coccoliths forming the spheres? This could be useful to put in the context of the dynamics of cell division.*

We agree that this would be useful information. However, it is difficult to unambiguously determine the number of layers of coccoliths under SEM. This would require a cross section of the cell, see e.g. Hoffmann et al. (2015), which is a method that was not available to us.

*(9) Page 7 Lines 35-36: The Authors should add a discussion on the mechanisms for this observation. There are a few studies on cells being stuck in the haploid phase due to the lack of N and P provision to replicate DNA and allow further division.*

Although the role of N and P in switching between haploid (non-calcified) and diploid (calcified) phases in coccolithophores is intriguing, to our knowledge no data has

been published so far to unequivocally show the role of nutrient limitation in phase switches. As we did not observe phase switches or address the haploid phase in the manuscript we feel that including this theory is too far removed from the main focus of this paper. We do discuss the lack of P-provision to replicate DNA, blocking

5      further cell division in the diploid phase, which is the reason for the cultures entering stationary phase under P-limitation.

*(10) Page 8 Lines 8-9: Please refer to recent study on this particular point by Aloisi in Biogeosciences 15: 4665-4692, and incorporate suitable discussion on the*

10    *mechanisms.*
We have elaborated on the mechanisms for the cell size increase under P-limitation based on the reference suggested by the reviewer and the work of Müller et al. (2008).

*(11) Page 9 Lines 6-7: I do not follow the argument being made here. Please clarify.*

15    The two possible reasons for the discrepancy between coccolith numbers per cell and cellular PIC content have now been separated into two paragraphs for clarity. The occurrence of partially dissolved coccoliths is now mentioned in the methods section instead of introducing it at this point.

20    *(12) Page 10 Lines 1-2: I recommend that the Authors tone this down, as we know that such a conclusion at the scale of the global biogeochemical cycle requires longer-term and multi-strain investigation although I appreciate the "may" being used here.*
This sentence has been reformulated.

**Anonymous Referee #3:**
*(1) They find that P-limitation can actually decrease the PIC quota, PIC production, and PIC/POC ratios in E. huxleyi, which is opposite that which has been most commonly reported in earlier literature, although some more recent studies are cited*

30    *to report similar results. This contrast is little discussed.*
Earlier experiments e.g. by Paasche used either batch or continuous (i.e. chemostat) cultures. As discussed in the text and illustrated by our data on P-limited batch cultures, there is an overriding effect of growth phase changes i.e. the change from exponential to stationary phase. In stationary phase, P-limitation blocks further cell

35    division and the cells remain in the G1 (assimilation) phase of the cell cycle in which both photosynthesis and calcification continue, leading to, respectively, an increase in POC content/cell size and an overcalcification of the cells.
A similar process may be at play in chemostats in which growth is continuous, but at a lower rate. Therefore cells will be in the assimilation phase longer which may lead

40    to an overproduction of coccoliths such as reported in Paasche and Brubak (1994). Please note that the POC content also increased in their chemostat. Similar results have been obtained by Müller et al. (2008) and Perrin et al. (2016). An exception is the study by Oviedo et al. (2014) who also reported a decrease in the PIC/POC ratio in five out of six *E. huxleyi* strains. However, also in this study the absolute PIC

content increased, together with POC content/cell size in stationary phase.

To our knowledge, ours is the first study to address nutrient limitation without introducing the confounding factor of growth rate changes. So most of this contrast can be explained by methodological differences. The relevant parts of the discussion
5   have been modified to more clearly illustrate this issue.

*(2) Neither the P-stress nor the heat stress used is very clearly justified. Where are such changes predicted to occur?*
P- and heat stress are expected to co-occur in a future warmer ocean. Our aim with
10  this study was to test the physiological limits of P-limitation and heat stress rather than testing specific predicted values.

*Why is P-stress chosen instead of N-stress, when much more of the world's ocean is thought to show N-limitation of primary production?*
15  Although much of the world's ocean is thought to be N- rather than P-limited, P-limitation can be relevant locally. This particular strain of *E. huxleyi* was isolated from the inner Oslo Fjord where the load of N over P can exceed the Redfield ratio and winter N concentrations are usually high (http://www.miljodirektoratet.no/old/klif/publikasjoner/2253/ta2253.pdf).
20  Furthermore, P-stress may be more relevant than N-stress for calcification as P-resources are necessary for energy storage and are a part of cellular membranes, two aspects that are relevant to coccolith production.

*(3) It is especially not clear to me what natural conditions are mimicked in the*
25  *P-limited batch cultures. Do E. huxleyi blooms naturally experience these chemical conditions (e.g., such low DIC and omega-calcite values)? If these are conditions arise in batch cultures at very high cell densities only reachable in lab monocultures, perhaps they must be more careful of extrapolating their results from stationary phase cultures to changes in carbon export.*
30  The reviewer correctly points out that the low DIC values (due to high cell concentrations) are unlikely to occur in nature. However, we used P-limited batch cultures vs. semi-continuous cultures to test physiological limits i.e. what effect does P-limitation have once it is growth-limiting i.e. blocking further cell division vs. when cell division is continuous. Furthermore, we only extrapolate results for carbon export
35  in the natural system from semi-continuous cultures (with more realistic cell concentrations and carbonate chemistry) and use P-limited batch cultures as a comparison.

*(4) In terms of heat stress, it's not made very clear why the temperatures of 19°C and*
40  *24°C were chosen, although there is some justification given in the Discussion. Is 19°C a typical SST in the North Sea (assuming the clone here represents a North Sea population) or typical of the Oslo Fjord? With global warming is it expected to reach 24°C regularly?*
A temperature of 19°C can be considered a high summer temperature in the Oslo

Fjord. As *E. huxleyi* often has maximum growth rates at temperatures above those found at the isolation site (Sett et al., 2014) we chose a temperature that was at the high end of the range that this strains is likely to encounter in nature as our "normal temperature".

5 It is unlikely that the Oslo Fjord will reach temperatures as high as 24°C regularly, at least on time scales that would preclude adaptation to higher temperature. The reason for choosing this temperature was that it was high enough to induce heat stress (defined as a decrease in growth rate) without causing the culture to crash. We were therefore focused on choosing a feasible culturing temperature that allowed us to
10 observe the physiological effects of heat stress rather than recreating the natural environment. We now mention the motivations/definitions for choosing these temperatures in the methods section.

*What about E. huxleyi populations currently found in 8-12°C waters, would the same*
15 *tendencies occur if grown at from 13°C to 19°C?*
This is a difficult question to answer and would depend very much on whether the physiological range of that strain corresponds to its natural range. As the reviewer rightly cites in the specific comments, numerous studies have tested the effect of temperature on carbon production in *E. huxleyi* using similar or higher temperatures
20 than we have (24°C). The temperature that is above the optimum will depend on the strain (and hence likely the place of isolation).
In this regard it is important to define "above-optimum temperature". This is defined as the temperature at which growth rate declines in respect to maximum growth rates obtained at the optimum temperature (Eppley, 1972). This is precisely why the
25 temperature of 24°C did not induce heat stress in Feng et al. (2008) as the growth rate increased from 20 to 24°C. Incidentally, this strain was isolated from the Sargasso sea (strain CCMP 371) with a known temperature range of 17-26°C. Rosas-Navarro et al. (2016) report that growth rates decreased in three strains of *E. huxleyi* at 27.5°C. However, they do not present data for PIC and POC production, or occurrence of
30 malformed coccoliths, at this above-optimum temperature, which is unfortunate as this would have been a very interesting data set for comparison.

*(5) It's not necessary for all studies to try to replicate specific environmental conditions (often impossible), perhaps especially when the goal is to understand*
35 *physiological limits or to take a first approximation.*
As the reviewer rightly points out, our choice of experimental set-up was based on testing the physiological limits of P-limitation and heat stress, which defined also the temperatures to use as outlined above.

40 *However, considering this lack of grounding of experimental design within an explicit environmental context, it appears that the Conclusions should be more cautious in extrapolating to biogeochemical effects.*
We have tried to outline our experimental design/motivations more clearly in the answers to the above comments and have also clarified these aspects in the text.

*(6) There is a focus on biogeochemical effects, but nothing on ecological effects of the documented changes in PIC and coccoliths. What function do they serve? I might suggest the review by Monteiro et al. to look at, and think of some of the consequences.*

A consideration of the possible functions of coccoliths and hence consequences of lower calcification rate/coccolith coverage has been added to the discussion.

*(7) Finally, I'm not so sure of the extent of the novelty of this study. They say "To our knowledge, this study is the first to specifically test the impact of heat stress…" but then there actually are a few quite relevant studies (it depends on how "heat stress" is defined), some of which they cite. For that reason, a more rigorous study design in an explicit environmental context would have been much stronger.*

We have now defined what we mean by the term "heat stress" i.e. a decrease in growth rate, both in the replies to the above comments and in the text. Considering this definition, we are not aware that other studies have examined the effect of heat stress on PIC production.

Specific comments:

*(1) p. 1 Line 26, should probably cite something more recent as well, such as the metaanalysis by Meyer and Riebesell 2015.*

This citation has been added.

*(2) p. 2 Lines 6-9 : "Batch culture on the other hand represents an end-of-bloom scenario in which the lack of nutrients limits further cell division… production cannot be determined in the batch approach ". That's only true if the last part of a batch culture is analyzed, as growth becomes limited due to exhaustion of nutrients, build-up of metabolites, shading, limited gas exchange, etc. In fact, there are many published experiments where production rates were determined in dilute batch culture, in the early exponential phase of growth before DIC consumption or nutrient consumption was substantial.*

The reviewer correctly points out that production rates can be determined in dilute batch cultures, in which growth is exponential. We could for example have examined the effect of heat stress in dilute batch cultures. However, it is not possible to test P-limitation in dilute batch cultures as these per definition will not be limited by nutrients such as P. We have modified this sentence to read that nutrient-limited production cannot be determined in the batch approach.

*(3) Line 29: "None of these studies, however, tested the effect of above-optimum temperature ". I don't understand this unless one defines what is "above-optimum temperature*

We have now defined "above-optimum temperature" upfront in the methods section.

*(4) Lines 36-37: Should consider (and cite) also work of van Bleiswijk et al. 1994 and*

*Rokitta et al. 2016 very relevant for theme of E. huxleyi response to P-limitation.*
The work of Wördenweber et al. 2017 is considered more relevant to this particular
study and has been included in this manuscript.

5   *(5) I have a problem with the use of K medium for nutrient experiments. K medium*
*contains a mix of ammonium and nitrate as N-source, and it contains*
*glycerophosphate as a P source. It's not clear from Gerecht et al. (2014) if they*
*modified these components. They need to give the basal medium composition they*
*used.*
10  It is correct that K-medium usually contains glycerophosphate as a P-source, as well
as ammonia. We, however, modified the medium to contain only nitrate as a N-source
and $KH_2PO_4$ as a P-source. The full medium recipe has been added to the methods
section (new table 1).

15  *What volume were cultures?*
Batch cultures were 350 mL, whereas semi-continuous cultures were kept at 50 mL
until the last dilution round where volumes were increased to 350 mL.

*(6) p.4 For semi-continuous cultures, what was the dilution rate or growth rate?*
20  The average growth rates for semi-continuous cultures are reported in Table 2. The
dilution rate varied slightly depending on the cell concentrations reached in the
cultures after two days as all cultures were diluted to 10.000 cells $mL^{-1}$ every two
days. A supplementary figure (1) has been added to show the development of cell
concentrations for each dilution cycle of semi-continuous cultures.

*How was it determined or confirmed that the cultures in fact were limited by P-*
*limitation in the semi-continuous cultures?*
During the course of the experiment we could not confirm whether the cultures were
limited, as the growth rate was not affected. We could confirm that the cultures were
30  limited only after harvesting the cultures and determining phosphorus (POP) content
which was lower in cultures grown on 0.5 μM initial phosphate medium than at 10
μM.

*How could maximum cell concentrations have reached 170000 cells/ml if cultures*
35  *were diluted back to 10000 cells/ml every second day? To have reached 170000*
*cells/ml from 10000 cells/ml in only 2 days would require aprox. 4 cell divisions per*
*day, which has never been reported for this species.*
To reach a maximum of 170.000 from 10.000 cells $mL^{-1}$ in two days requires approx.
4 cell divisions in two days i.e. approx. 2 cell divisions per day, which is in the range
40  of what has been reported for this species.

*(7) Line 7: "P-limited cultures were harvested in stationary phase, ..." for how long*
*in stationary phase? This is not clear from Fig. 4.*
P-limited cultures were harvested on the day of the last data point presented in Fig. 4

(new Fig. 1). We have added arrows to the figure to illustrate this.

*(8) p. 5 Line 10: Give manufacturer & city for "CASY".*
Manufacturer & city are given at first mention of the instrument (p. 4 line 14).

*(9) Line 30: "Average values were compared by a t-test". Was this pairwise test performed after the two-way ANOVA? If so, with what correction for multiple comparison? They are testing two factors (T and P-limitation) so should be doing a two-way ANOVA, not t-tests.*

10 The reviewer rightly points out that a two-way ANOVA is necessary to test the effect of two factors (T and P-limitation). We used t-tests when comparing the effect of only one of these factors. However, as we describe both factors together in the text, the reference to the t-test is obsolete and has been removed. As stated in the next sentence, we used a two-way ANOVA to compare the data.

*(10) p. 6 Line 24 and Table 3: What limited the growth of control batch cultures?*
Nothing was limiting control batch cultures as they were harvested in exponential phase.

20 *(11) p. 8 Lines 10-12: "These large "ready-to-divide" cells (Gibbs et al., 2013) not only accumulate POC, but also accumulate PIC, leading to the 2-3-fold increase in coccolith number per cell observed in stationary phase cultures (Fig. 2c,d)." How do you know these cells are "ready-to-divide"? If they really are "ready-to-divide", do you mean they are blocked in G2 or M phase of the cell cycle? That doesn't make*
25 *much sense.*
This sentence has been rephrased.

*(12) Lines 10-12: This is an important justification for their selection of temperatures. Nevertheless, I'm not very convinced about how these temperatures aare reoe. I*
30 *would prefer them to explicitly give the range of temperatures experienced in the North Sea as well as the fjord. Why is 19 °C a "normal temperature"? What does that mean?*
The natural temperature range that can be expected for this strain has been added to the text.

*(13) Lines 12-13: "Stationary phase can be likened to an end-of-bloom scenario in nature, during which E. huxleyi sheds numerous coccoliths, leading to the characteristic milky color of coccolithophore blooms". Maybe, but it's also well know that the end of E. huxleyi blooms involves infection by the virus EHV.*
40 In this statement we are not referring to what causes the demise of *E. huxleyi* blooms in nature (for which there may be numerous reasons), but rather describing the characteristic "overproduction" of coccoliths.

*(14) p. 9 Line 10 "The percentage of partially dissolved coccoliths was higher at*

*normal temperature than under heat" Where is this shown? Data is presented on*
*"incomplete", "malformed", and "normal" coccoliths in Fig. 3. They state "The high*
*numbers of incomplete coccoliths observed in P-limited batch cultures were likely a*
*result of secondary dissolution (Fig. 1d; Langer et al., 2007) due to the low calcite*
5  *saturation state reached in stationary phase cultures." I would like to see more*
*examples of incomplete coccoliths. Perhaps they can show that the type of*
*incompleteness that appears in P-limited batch cultures (when omega-calcite is less*
*than 1) is distinct from what appears when omega-calcite is greater than one?*
Our data set is complicated by the occurrence of high amounts of dissolved coccoliths
10  in P-limited batch cultures, the features of which we describe in a new table in the
methods section. This dissolution affects all categories of coccoliths (normal,
malformed, incomplete). However, the features of dissolved coccoliths are similar to
those of incomplete coccoliths and we were not able to make an unambiguous
distinction between the two possible origins of "incomplete coccolith morphology"
15  i.e. "incompletely produced" or "incomplete because of secondary dissolution". We
therefore only have one category of "incomplete coccoliths" in our Fig. 3, most of
which are a result of dissolution in P-limited stationary phase batch. We have added
an upfront description of these differences in the methods section as well as additional
images to (new) Fig. 2.

**Anonymous referee #4:**
*(1) I feel that the discussion is weakened by an emphasis on comparing two culturing*
*methods rather than comparing individual vs. interactive effects of the stressors in*
*question.*
25  We would like to point out that the inclusion and discussion of the two culturing
methods in the manuscript not only serves as a "methodological comparison". It also
serves to compare two differing environmental scenarios. Whereas batch culture
represents a strong P-limitation as may be encountered at the end of blooms, the semi-
continuous culture tests the effect of a more continuous low-P environment.
30  We agree that the manuscript would benefit from a comparison of individual vs.
interactive effects of the two stressors and a short discussion has been added to the
conclusions section.

*(2) The specific choice of experimental conditions is also poorly justified and not*
35  *placed into context.*
The choice of the experimental conditions is now described in the methods section.

*Specific Comments:*
*(1) In the broader interpretations of their calcification results, the authors state in the*
40  *Abstract, Introduction and Conclusions that decreases in calcification rates in E.*
*huxleyi could "lessen the ballasting effect of coccoliths and weaken carbon export out*
*of the photic zone", or similar wording. In support, they reference Ziveri et al. (2007),*
*who conclude that, despite the high abundance of Emiliania relative to other*
*coccolithophore species, the small size and very low species-specific carbonate mass*
45  *of their coccoliths means that they consequently export far less carbonate than*
*expected. Baumann et al. (2004) similarly concluded that Emiliania plays only a*
*relatively minor role in carbonate export in the Equatorial and South Atlantic.*

*Can the authors support their statements of alterations to the carbon cycle more quantitatively using their PIC production values and abundances of Emiliania in the field? Or provide references of studies that better support these statements?*

The reviewer correctly points out that the overall contribution of *E. huxleyi* to pelagic carbonate flux is small compared to other species. Nevertheless, coccolith ballasting (or lack thereof) can be considered relevant locally e.g. during blooms of *E. huxleyi*. To support the above statement i.e. weaker ballasting due to decreased calcification rates, it is necessary to examine whether and to what extent these physiological responses are applicable to coccolithophores as a whole. We have some indication that other species react similarly, including those arguably more (regionally) relevant to carbonate export such as *C. pelagicus*. For example, we have previously reported that *C. pelagicus* increases P-requirements and decreases coccolith coverage under heat stress (Gerecht et al., 2014). The relevant parts of the text have been modified to reflect this.

*(2) Considerable emphasis is made on the short-comings of the batch culturing technique compared to the semi-continuous culture technique. Comparison of these two methods is present throughout the results and discussion, and, in my opinion, obscures a clear and explicit evaluation of contrasting individual (warming only, P-limitation only) vs. interactive (warming and P-limitation) effects and evidence (or not) for positive interactions.*

Please see reply to general comment (1).

*It is stated that batch culture experiments can only represent a severely nutrient depleted scenario whereas a semi-continuous set-up provides an acclimated low-nutrient population (p3, ln 7; p7 lns 12-17). I think that these statements are somewhat misleading for the following reason: A batch culture experiment experiences exponential growth at whatever the starting nutrient concentrations until these nutrients become sufficiently depleted that exponential rates of growth can no longer be maintained and growth rate rapidly falls to zero. A semi-continuous culture is just a batch culture that is subcultured/diluted (typically) around mid-exponential-phase cell concentrations several times. I therefore find it strange that the authors did not just sample their P-limited batch culture experiment at mid-exponential phase (as they did with the control experiments) well before the 'severe' nutrient depletion of stationary phase began, which based on Fig. 4 would have meant sampling on day 3 or 4. Would this not have been a more realistic comparison of control and P-limited conditions during exponential phase in all experiments?*

Although the reviewer is correct insofar as there are similarities between an exponential batch culture and a semi-continuous culture, these two scenarios are not identical. The crucial difference is that a semi-continuous culture experiences stable limiting conditions over many generations, which leaves ample time for acclimation processes to re-structure the physiological machinery dealing with this environmental stress. In a batch culture, by contrast, the specific state of limitation equivalent to a semi-continuous scenario is a transient state experienced by the cells for a short time only (less than one generation). The whole point of comparing batch and semi-continuous culture was to compare the cumulative effect of a series of transient limitation states (increasingly severe; batch) to a single constant limitation state representing roughly the average of the many transient batch culture states (semicontinuous). Our experimental setup was designed to serve this purpose.

*(3) I would like to see a broader context of the area where a similar degree of temperature and phosphorus stress is predicted to be experienced in the context of this strain isolated from a Norwegian fjord.*

The aim of this study was not to test/evaluate specific predicted environmental factors relevant for our strain. The aim was to look at environmental stressors (high temperature, P-limitation) by testing the physiological limits, independently of whether and when this strain will encounter these conditions in nature. Therefore we chose a temperature that was above the optimum for growth (but that still allowed growth) to test the effect of heat stress. Similarly we tested the effect of strong and weak P-limitation on this strain by having one laboratory scenario in which P-limitation becomes limiting for cell replication (stationary phase batch culture) and one in which P-limitation is not strong enough to affect cell replication i.e. growth rate, but does affect calcification rate (semi-continuous culture).

*Similarly, there is no discussion of the fact that physiological stress experience by one strain of this species under climate change is as likely to lead to its ecological replacement by another, more tolerant strain given recent studies presenting the large genetic pool of Emiliania (e.g., Read et al., 2013, which also discusses differences in genes for tolerance of low phosphorus conditions between strains).*

This aspect has now been considered in the discussion.

*(4) I was surprised to see that no figures of any POC, PIC, or POP data were presented, only data in the tables. Was there a reason for this? Given the two experimental approaches, two temperatures and two nutrient states, it made it difficult to quickly visualise the dataset.*

There was no particular reason, apart from keeping the manuscript concise. The relevant figures have been added as a supplement to the final version (new supplementary Fig 2.).

*(5) The strain used seems to be a new isolate – do the authors intend to deposit this strain into a culture collection for use by other researchers? Given it is not held in a culture collection, the authors must provide the essential ancillary information on the isolate and its maintenance in culture.*

This has been added.

*(6) It is stated deep into the discussion that the temperature at the isolation location does not exceed 21 degrees and this is presumably how the authors decided that a temperature of 24 degrees was beyond the thermal optimum. Did the authors perform a systematic temperature optimum assessment by determining growth rates at a range of temperatures?*

We agree that it would have been beneficial to rigorously test growth rates over a broad range of temperatures before choosing the applied temperatures of 19 and 24°C. The fact that we did not do this, however, does not detract from the fact that 24°C was above the optimum for growth. We have mentioned the motivations/definitions for choosing the two temperatures upfront in the methods section.

*Given that the exponential growth rates were not substantially different between*

*temperature treatments (semi-continuous) and in fact were higher in the 24 degree*
*treatment for the batch culture approach, this would perhaps suggest that (using*
*exponential growth rate as a physiological indicator) this isolate has a relatively*
*broad thermal tolerance.*

5    A decrease in growth rate, even slight, at a higher temperature is the definition of
above-optimum growth. According to Eppley (1972), phytoplankton growth rates
increase with increasing temperature as long as this temperature is below the optimum
for growth, whereas above optimum temperature, growth rates decline. As growth
rates decline more sharply at above-optimum temperatures (i.e. heat stress is more

10    detrimental than below-optimum temperature), it is often difficult to culture
phytoplankton at above-optimum temperature. Therefore, obtaining a stronger
difference in growth rate at even higher temperature would have been technically
difficult and would likely have resulted in the crash of the culture.

15    *(7) The authors do not state whether there was any period of acclimation for*
*populations experiencing low phosphate or high temperature treatments.*
Cultures were acclimated for ca. 10 generations to low phosphate and high
temperature culture conditions before starting the experiment. This information has
been added to the methods section.

   *(8) How did the authors account for the tendency of Emiliania to form multi-layer*
*coccospheres when counting the number of coccoliths from SEM images? Comparing*
*the coccosphere size from CASY with the cell size from light microscopy (back*
*calculated from the volume data) and considering the thickness of Emiliania*

25    *coccoliths, would suggest that coccospheres were not mono-layer.*
The majority of multilayered coccospheres found in our study collapsed during the
filtration process, which allowed us to count the coccoliths from all layers. The non-
collapsed multilayered coccospheres commonly had an only partially covered first
layer i.e. an incomplete second layer. Thus, it was possible to count the visible

30    coccoliths in the first and second layer and estimate the number of covered coccoliths
of the first layer. In very rare cases when we could not estimate the number of layers
due to a complete outer layer, we estimated the number of layers and the number of
coccoliths, which could fit under the outermost layer. This approach was consistently
used throughout the SEM analysis to minimize error arising from the inability to see

35    all coccoliths on the coccosphere.

   *(9) Is there any reason why the authors refer to both light microscope and CASY cell*
*size measurements as "cell size" when CASY gives coccosphere size measurements?*
*There are huge differences in volumes between the two methods due to the*

40    *coccosphere and whilst cell size directly relates to cell carbon, coccosphere size does*
*not and therefore this unnecessarily confuses the reading in parts.*
The CASY system does not actually provide accurate coccosphere size measurements
(Gerecht et al., 2015), but gives an intermediate value between cell and coccosphere
size. We therefore use CASY measurements as a proxy for "cell size" to be able to

45    observe and visualize the size increase during the development of the batch culture.

   *(10) The discussion would benefit from a discussion of the physiological mechanisms*
*behind the observed response to P-limitation and heat stress singularly and*
*combined, and there is considerable literature on this species, other coccolithophore*

*species and other phytoplankton groups that would support such a discussion.*
We do discuss the possible physiological mechanism behind the cell size increase under P-limitation, as well as the possible explanations for the increased P-requirements under heat stress. As this is the first observation of decreased calcification rate under heat stress and P-limitation in this species, we find it difficult to speculate further about the possible physiological mechanisms unless the referee would like to point us towards relevant literature. In this regard, please also see reply to general comment (4) of referee #2.

*p6, ln 37-39 – referred to changes in size but "twice as large in stationary phase" refers to cell volume, so this should be changed to reflect this.*
The reference to cell volume has been added.

*p8, ln 7-9 – The reference to Sheward et al. (2016) on line 7 should be changed to Gibbs et al. (2013) as the latter presented Emiliania data. This sentence could also include C. braarudii, Calcidiscus and Helicosphaera from Sheward et al. (2017). The Sheward et al. (2016) at the end of line 9 should be changed to 2017 (I think you have referred to the discussion paper rather than the finally-published article).*
The relevant references have been changed.

*Technical Corrections:*
*Throughout the paper, there are inconsistencies with the author order in your references. Sometimes they are ordered by date, other times alphabetically, and several times I can find no logic to the order! (e.g., p2, ln 37-38).*
This has been resolved.

**Anonymous referee #5:**
*...the choice of the cultures conditions, P limitation and temperature stress, need to be express with more details in the introduction or in the methodology part concerning the choice of the values.*
This has been added.

*Specific comments*
*p. 2; Ln 9: You need to add the word particulate to present for the first time in your manuscript the PIC and POC terms.*
This has been added.

*p. 3; 19: What does inter alia mean?*
Latin for "among other things"; "inter alia" has been replaced with this term in the text.

*p. 3; Ln 29 & 35 : Did you test different temperature on the strain (before these experiment) to find the optimal temperature at the selected light dose (12 :12, 100_mol photons.m-2.s-1)? Or did you have an idea (or reference) of the optimal temperature of your strain before selecting your two temperature conditions?*
We agree that it would have been beneficial to rigorously test growth rates over a broad range of temperatures before choosing the applied temperatures of 19 and 24°C. The fact that we did not do this, however, does not detract from the fact that 24°C was above the optimum for growth. We choose these temperatures based on the natural temperature range for this strain. This motivation has been added to the text.

*p.3; Ln 34-39: What is the name of your strain?*
Please see reply to general comment (1) of referee #2.

*Have you done some acclimations of your strain to the temperature and P*
*concentration conditions before starting your experiment? You should indicate in*
*your method if you did.*
Yes, cultures were acclimated for ca. 10 generations to the temperature and P
concentration conditions before starting the experiment. This information has been
added to the methods section.

*What is the initial nitrate concentration in your medium? Is it the standard*
*concentration of NO3 in a K medium? If not, the value of the concentration need to be*
*indicate in this method part.*
The basal composition of the medium has been added to the methods section.

*You also need to add the value of the salinity of your medium.*
This information has been added to the methods section (new table 1).

*p. 5; Ln 21-26: Did you take into account that Emiliania huxleyi can have multiple*
*layers of coccoliths? It is clearly visible on your figure 1 that your strain can have*
*several layers of coccoliths depending of the cultures conditions. How did you take*
*this particularity in your coccoliths counts?*
Please see reply to specific comment (8) of referee #4.

*What is the standard deviation on your counts? At least, you have triplicate so you*
*need to specify the standard deviation on your number.*
The standard deviation of our counts of coccolith number per cell is listed for each
culture condition in (the new) table 3. This standard deviation is based on the total
number of cells analysed (n), which is the sum of all three replicate cultures for each
condition. N is listed for each culture condition in table 3.

*What about coccospheres diameter on your SEM images? Did you think about doing*
*coccopheres measurements? If not, is it because of the high number of detached*
*coccoliths on your filter?*
As the reviewer points out, there was a high number of detached coccoliths on the
filter so that coccosphere measurements under SEM would possibly have been
underestimated. Additionally, most of the coccospheres in P-limited batch cultures
had collapsed, making accurate estimates of coccosphere size based on SEM
measurements difficult.

*It should be interesting to discuss the PIC content related to your coccolith number*
*and the thickness of your coccolith layer.*
Yes, we agree with the reviewer that this would have been an interesting aspect to
examine. However, determining the thickness of the coccolith layer or the number of
coccolith layers around one cell with SEM was not possible; see also response to
specific comment (8) of referee #2.

*p. 6; Ln 24-34: Final cell density for batch cultures were being push really high in*
*order to get the P-limitation. However, respect to LaRoche et al. 2010, inorganic*
*carbon system changes need to be kept below 5% to avoid carbon system changing*

*due to the high cell density. This meens that nutrient limitation experiment for batch system have to attain a reasonable final cell density in order to keep a DIC system quasiconstant through the experiment and to avoid a inorganic carbon limitation before the P limitation in this case. In your P-limited cultures, DIC changes that you* 5 *get at the end of the experiment led to a low calcite saturation constant. Therefore observations in morphology and calcite content may be due to this changes rather than the P-limitation and heat stress effect. You took into account this changes in your discussion but it will be useful to indicate why you did batch experiments in this way. Could you justify this choice in the methodology of your batch experiment? Did you* 10 *think about carrying the experiment with a lower initial P concentration rather than push the cell density so high? Or did you have a target initial P concentration that you wanted to test? In this last case, a comment on the 2.1 part will be necessary.*

We did not choose an initial phosphate concentration of 0.5 μM to recreate a specific field situation. Rather, this concentration was chosen based on prior considerations of collecting enough biomass for

15 all analyses, while keeping cell concentrations of semi-continuous cultures well below stationary phase. The low DIC concentrations in P-limited stationary phase batch cultures do no affect the conclusions of the manuscript in regard to P-limitation, which are that P-limitation does not affect coccolith morphology per se, but decreases calcification rate in this strain of *E. huxleyi*.

20 *p. 7; Ln 27: What is the normal temperature? Even if you explain it later (p.8 Ln 26), you should describe here that the normal temperature is 19°C if you want to use this term. It is not obvious for readers.*

The terms (and motivations) of using 19°C as normal and 24°C as supraoptimal temperature have been now explained in the methods section.

*p. 8; Ln 1-5: same comments that previously on the DIC system*

See response to previous comments.

*Figures & Tables*

30 *Figure 2: You need to add the color explanations: blue is your schematic initial cell, red is the schematic effect observed at the end of your experiment.*

This information has been added.

*Figure 4: You should add the error bar on triplicate somewhere in your legend.*

35 This information has been added.

*Tables 1, 2 & 3: You should add the standard deviation in your legend and your n number (for example, n = 3 if it is triplicate).*

This has been added.

*Table 2: You need to clarify in the legend of your table if the cell volume has been calculated with measurements of cell diameter from the harvest day or from an average of daily measurements.*

The cell volume presented in the table was calculated from measurements of cell

45 diameter from the harvest day. This information has been added to the table.

Cited references

Aloisi, G.: Covariation of metabolic rates and cell size in coccolithophores, Biogeosciences, 12, 4665-4692, 2015.

Borchard, C., Borges, A. V., Händel, N., and Engel, A.: Biogeochemical response of *Emiliania huxleyi* (PML B92/11) to elevated $CO_2$ and temperature under phosphorus limitation: a chemostat study, J. Exp. Mar. Biol. Ecol., 410, 61-71, 2011.

Buitenhuis, E. T., de Baar, H. J. W., and Veldhuis, M. J. W.: Photosynthesis and calcification by

5 *Emiliania huxleyi* (Prymensiophyceae) as a function of inorganic carbon species, J. Phycol., 35, 949-959, 1999.

[revised manuscript text omitted]

30 This study investigates the physiological and morphological response of *E. huxleyi* to two environmental stressors, phosphorus (P) limitation and increased temperature. These are predicted to occur simultaneously as a rise in global temperature will increase the likelihood of nutrient limitation in the photic zone due to a stronger stratification of the water column (Sarmiento et al., 2004). The availability of macronutrients such as nitrogen and P have been shown to affect the production of

35 particulate organic (POC) and inorganic carbon (PIC) in coccolithophores (reviewed by Zondervan 2007). Coccolith number per cell generally increases in P-limited cultures, often leading to an increase in the PIC/POC ratio (Paasche and Brubak, 1994; Paasche, 1998; Müller et al., 2008; Perrin et al., 2016). However, five out of six Mediterranean *E. huxleyi* strains showed a decreased PIC/POC ratio in response to P-limitation, and one strain displayed no change (Oviedo et al., 2014). While this

40 demonstrates that there are strain-specific responses to P-limitation, some differences between studies
* * *
Information Services 29/9/2017 12:45

Information Services 21/9/2017 08:50

Information Services 25/9/2017 09:24

Information Services 21/6/2017 13:19

Information Services 25/9/2017 09:25

Information Services 25/9/2017 09:29

Information Services 25/9/2017 09:27

Information Services 25/9/2017 09:27

on PIC and POC production are due to differences in experimental methods, notably batch culture and (semi)-continuous culture approaches (Langer et al., 2013b). We used both set-ups in this study to examine the difference between strong, yet brief P-limitation (stationary phase batch culture) against weak, but continuous P-limitation (semi-continuous culture). The latter method best represents areas with permanently low nutrient availability such as the eastern Mediterranean (Krom et al., 1991; Kress et al., 2005), while stationary phase batch culture can be approximated to an end-of-bloom scenario in which the lack of nutrients limits further cell division. Both approaches are relevant in ecological terms, but for methodological reasons (i.e. non-constant growth rates), nutrient-limited production cannot be determined in the batch approach (e.g. Müller et al., 2008; Langer et al., 2012; Langer et al., 2013b, Gerecht et al., 2014; Oviedo et al., 2014; Perrin et al., 2016). In a (semi)-continuous culturing set-up, growth rate is constant over the course of the experiment and production rates can be calculated (Paasche and Brubak, 1994; Paasche, 1998; Riegman et al., 2000; Borchard et al., 2011). Ratio data such as coccolith morphology, on the other hand, should be comparable between batch and (semi)-continuous culture experiments (Langer et al., 2013b) as has been shown for *C. pelagicus* (Gerecht et al., 2014; Gerecht et al., 2015). However, the only strain of *E. huxleyi* (B92/11) that was tested in both batch and continuous culture was not analyzed for coccolith morphology and the PIC/POC ratio showed a markedly different response to P-limitation in batch and in continuous culture (Borchard et al., 2011; Langer et al., 2013b). In this study we therefore tested another strain of *E. huxleyi* in both semi-continuous and batch culture and analyzed among other things, coccolith morphology and the PIC/POC ratio.

[revised manuscript text omitted]

Information Services 25/9/2017 10:20